# Hyperbolic VAE via Latent Gaussian Distributions

**Seunghyuk Cho**
POSTECH GSAI
shhj1998@postech.ac.kr

**Juyong Lee**
KAIST AI
agi.is@kaist.ac.kr

**Dongwoo Kim**
POSTECH GSAI & CSED
dongwoo.kim@postech.ac.kr

## Abstract

We propose a Gaussian manifold variational auto-encoder (GM-VAE) whose latent space consists of a set of Gaussian distributions. It is known that the set of the univariate Gaussian distributions with the Fisher information metric form a hyperbolic space, which we call a Gaussian manifold. To learn the VAE endowed with the Gaussian manifolds, we propose a pseudo-Gaussian manifold normal distribution based on the Kullback-Leibler divergence, a local approximation of the squared Fisher-Rao distance, to define a density over the latent space. We demonstrate the efficacy of GM-VAE on two different tasks: density estimation of image datasets and state representation learning for model-based reinforcement learning. GM-VAE outperforms the other variants of hyperbolic- and Euclidean-VAEs on density estimation tasks and shows competitive performance in model-based reinforcement learning. We observe that our model provides strong numerical stability, addressing a common limitation reported in previous hyperbolic-VAEs. The implementation is available at `https://github.com/ml-postech/GM-VAE`.

## 1 Introduction

The geometry of latent space in generative models, such as variational auto-encoders (VAE) (Kingma & Welling, 2013), reflects the structure of the data representations. Mathieu et al. (2019); Nagano et al. (2019); Cho et al. (2022) show that employing hyperbolic space as the latent space improves the preservation of the hierarchical structure within the data. The theoretical background for adopting hyperbolic space lies in the analysis of Sarkar (2011); the tree-structured data can be embedded with arbitrary low distortion in hyperbolic space, while Euclidean space requires extensive dimensions.

Previously proposed hyperbolic VAEs rely on Poincaré normal distribution (Mathieu et al., 2019) or hyperbolic wrapped normal distribution Nagano et al. (2019) for the prior and variational distributions. Unlike the Gaussian distribution in Euclidean space, however, these distributions suffer from several shortcomings, including the absence of closed-form Kullback-Leibler (KL) divergence, numerical instability (Mathieu et al., 2019; Skopek et al., 2019), and high computational cost in sampling (Mathieu et al., 2019).

Meanwhile, we can form a Riemannian manifold from the set of univariate Gaussian distributions by equipping the Fisher information metric (FIM). It is known that the FIM of univariate Gaussian distributions is akin to that of the metric tensor of the Poincaré half-plane model (Costa et al., 2015), providing a perspective of viewing the points in hyperbolic space as univariate Gaussian distributions. In other words, a Gaussian distribution can be mapped to a single point in the open half-plane manifold as shown in Figure 1, where the FIM forms the shortest geodesic distance between two Gaussian distributions. Noting that the numerical issue of Poincaré normal arises from the geodesic distance of hyperbolic space, we question whether this perspective can lead us to define a new distribution with better analytic properties.

In this work, inspired by the fact that KL divergence itself is a statistical distance that locally approximates the geodesic distance (Tifrea et al., 2018), we propose a hyperbolic distribution by

37th Conference on Neural Information Processing Systems (NeurIPS 2023).

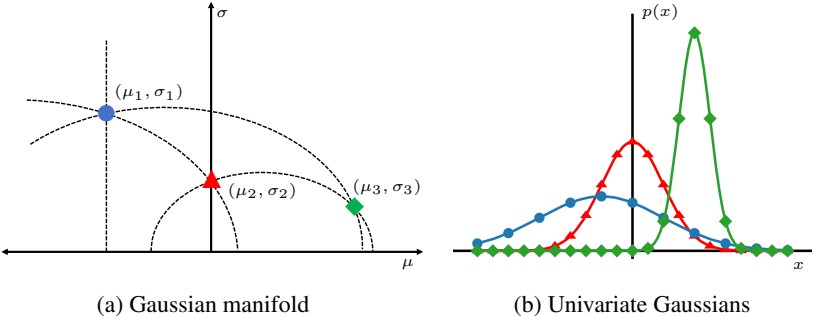

(a) Gaussian manifold          (b) Univariate Gaussians

Figure 1: (a) The visualization of the Gaussian manifold consisting of a set of Gaussian distributions. Each point of the Gaussian manifold is a pair of two parameters of a univariate Gaussian distribution: $(\mu, \sigma) \in \mathbb{R} \times \mathbb{R}_{>0}$. The dashed lines are the geodesics, which are either the ellipses with eccentricity $1/\sqrt{2}$ with the origin placed on the $\mu$-axis or straight lines parallel to the $\sigma$-axis. (b) Three univariate Gaussian distributions correspond to three points in the Gaussian manifold in (a).

substituting the geodesic distance of Poincaré normal with the KL divergence between the univariate Gaussian distributions. We then verify that this simple yet powerful alteration results in several practical analytic properties; the proposed distribution reduces into the product of two well-known distributions, i.e., the Gaussian and gamma distributions, which are easy to sample, with a closed-form KL divergence between the proposed distributions. By adopting the proposed hyperbolic distribution, we introduce a new variant of hyperbolic VAE, named Gaussian manifold VAE (GM-VAE), whose latent space is a set of Gaussian distributions.

During the experiments, we observe that the proposed distribution is robust in terms of sampling and KL divergence computation compared to the commonly-used hyperbolic distributions; we briefly explain the reason why others are numerically unstable. Experimental results on the density estimation task with image datasets show that GM-VAE can achieve outperforming generalization performances to unseen data against baselines of Euclidean and hyperbolic VAEs. Application of GM-VAE on model-based reinforcement learning (RL) verifies the feasibility of using hyperbolic space on another domain of task.

We summarize our contributions as follows:

- We introduce a variant of VAE whose latent space is defined on a statistical manifold formed by univariate Gaussian distributions, namely Gaussian manifold.

- We propose a new distribution, called a pseudo Gaussian manifold normal distribution, which is easy to sample and has closed-form KL divergence, to train VAE on the Gaussian manifold.

- We empirically verify that the newly proposed VAE performs stable training without numerical issues on the density estimation task with several image datasets. The proposed model outperforms the baseline Euclidean VAE and other hyperbolic variants.

- We show that our method can be used for model-based RL. Specifically, we replace the latent space of the world model with hyperbolic space for learning environments, showing competitive results with a state-of-the-art baseline.

## 2 Preliminaries

In this section, we first review the fundamental concepts of hyperbolic space and commonly used hyperbolic models. We then explain the Riemannian geometry between statistical objects, showing the connection between the statistical manifold and hyperbolic space.

### 2.1 Review on hyperbolic space

**Riemannian manifold.** A $n$-dimensional Riemannian manifold consists of a manifold $\mathcal{M}$ and a metric tensor $g : \mathcal{M} \to \mathbb{R}^{n \times n}$, which is a smooth map from each point $\mathbf{x} \in \mathcal{M}$ to a symmetric

positive definite matrix. The metric tensor $g(\mathbf{x})$ defines the inner product of two tangent vectors for each point of the manifold $\langle \cdot, \cdot \rangle_{\mathbf{x}} : \mathcal{T}_{\mathbf{x}}\mathcal{M} \times \mathcal{T}_{\mathbf{x}}\mathcal{M} \to \mathbb{R}$, where $\mathcal{T}_{\mathbf{x}}\mathcal{M}$ is the tangent space of $\mathbf{x}$.

The metric tensor induces basic Riemannian operations, such as a geodesic, exponential map, log map, and parallel transport. Given two points $\mathbf{x}, \mathbf{y} \in \mathcal{M}$, geodesic $\gamma_{\mathbf{x}} : [0, 1] \to \mathcal{M}$ is a unit speed curve on $\mathcal{M}$ being the shortest path between $\gamma(0) = \mathbf{x}$ and $\gamma(1) = \mathbf{y}$. This curve can be interpreted as a generalized path of a straight line in Euclidean space. The exponential map $\exp_{\mathbf{x}} : \mathcal{T}_{\mathbf{x}}\mathcal{M} \to \mathcal{M}$ is defined as $\exp_{\mathbf{x}}(\mathbf{v}) = \gamma(1) = \mathbf{y}$ given $\gamma$ is a geodesic starting from $\gamma(0) = \mathbf{x}$ and $\gamma'(0) = \mathbf{v} \in \mathcal{T}_{\mathbf{x}}\mathcal{M}$. The log map $\log_{\mathbf{x}} : \mathcal{M} \to \mathcal{T}_{\mathbf{x}}\mathcal{M}$ is the inverse of the exponential map, i.e., $\log_{\mathbf{x}}(\exp_{\mathbf{x}}(\mathbf{v})) = \mathbf{v}$. The parallel transport $\mathrm{PT}_{\mathbf{x} \to \mathbf{y}} : \mathcal{T}_{\mathbf{x}}\mathcal{M} \to \mathcal{T}_{\mathbf{y}}\mathcal{M}$ moves the tangent vector $\mathbf{v}$ along the geodesic between $\mathbf{x}$ and $\mathbf{y}$. The geodesic distance $d_{\mathcal{M}}(\mathbf{x}, \mathbf{y})$ can be induced by the metric tensor as follows:

$$d_{\mathcal{M}}(\mathbf{x}, \mathbf{y}) = \int_0^1 \sqrt{\langle \dot{\gamma}(t), \dot{\gamma}(t) \rangle_{\gamma(t)}} dt.$$

**Hyperbolic space.**    One method of classifying Riemannian manifolds, a basic question of differential geometry, is based on curvature. Among different types of curvatures, one popular curvature is the sectional curvature $\kappa_g$, which is a generalization of the Gaussian curvature in classical surface geometry. Given two linearly independent vector fields $X, Y \in \mathfrak{X}(\mathcal{M})$, the sectional curvature $\kappa_g$ can be computed with Riemannian curvature tensor $R \colon \mathfrak{X}(\mathcal{M}) \times \mathfrak{X}(\mathcal{M}) \times \mathfrak{X}(\mathcal{M}) \to \mathfrak{X}(\mathcal{M})$ as below:

$$\kappa_g = \frac{\langle R(X, Y)Y, X \rangle}{\langle X, X \rangle \langle Y, Y \rangle - \langle X, Y \rangle^2},$$

where $R(X, Y)Y$ returns a tensor field assigning a tensor to each point of the Riemannian manifold $\mathcal{M}$. The hyperbolic space is a Riemannian manifold that has the sectional curvature value of constant negative (Nickel & Kiela, 2018). The hyperbolic space is known to be able to embed tree-structured data with arbitrarily low distortion (Sarkar, 2011).

**Hyperbolic models.**    We utilize three famous models of hyperbolic space: the Poincaré disk model, the Lorentz model, and the Poincaré half-plane model.

The Poincaré disk model is a hyperbolic space with an open disk manifold. Earlier hyperbolic machine learning work uses the Poincaré disk model because it has a simple closed-form of the operations, such as exponential and log maps (Mathieu et al., 2019). However, the Poincaré disk model suffers from numerical stability issues when the points exist near the boundary of the manifold.

The Lorentz model is often used as an alteration of the Poincaré disk model (Nickel & Kiela, 2018; Nagano et al., 2019; Bose et al., 2020; Cho et al., 2022). The Lorentz model uses a half hyperboloid manifold, where the closed form of the Riemannian operations exists, so the numerical stability issue of employing the Poincaré disk model is relieved.

The Poincaré half-plane model is another well-known model of hyperbolic space with an open half-plane manifold. The metric tensor of a point of the two-dimensional Poincaré half-plane model $(x, y)$ is $y^{-2}\mathrm{diag}(1, 1)$.

**Numerical stability issues of the hyperbolic models.**    Hyperbolic space suffers from numerical stability when applied to machine learning algorithms (Yu & Sa, 2021; Skopek et al., 2019; Mathieu et al., 2019). The numerical stability mainly occurs for two reasons: machine precision error and unstable Riemannian operations.

First, due to the machine precision error, the hyperbolic points represented with floating point differ from the real value (Yu & Sa, 2021, 2019). In contrast to Euclidean space, the points of hyperbolic space need to satisfy manifold constraints, e.g., the Poincaré disk model allows points whose Euclidean norm is less than one. A point can be placed near the boundary during the optimization or inference processes. Although the point does not violate the manifold constraint in theory, it can be located on or out of the boundary when represented with a floating point due to machine precision error. We empirically observe that the manifold constraint violation occurs frequently when we need to embed many data points in hyperbolic space. Figure 3 demonstrates the machine precision error of each hyperbolic model.

Second, the Riemannian operations of hyperbolic space can result in a not-a-number (NaN) value when the input value is not in the manifold. For example, the geodesic distance from the Poincaré

disk model and the log mapping of the Lorentz model are unstable Riemannian operations, which are written as:

$$d_{\mathcal{P}}(\mathbf{x}, \mathbf{y}) = \cosh^{-1}\left(1 + 2\frac{\|\mathbf{x} - \mathbf{y}\|^2}{(1 - \|\mathbf{x}\|^2)(1 - \|\mathbf{y}\|^2)}\right), \log_{\mathbf{u}}(\mathbf{v}) = \frac{\cosh^{-1}(\alpha)}{\sqrt{\alpha^2 - 1}}(\mathbf{v} - \alpha\mathbf{u}),$$

where $\alpha = \mathbf{u}_0\mathbf{v}_0 - \sum_{i=1}^n \mathbf{u}_i\mathbf{v}_i$ is the Lorentzian inner product of $\mathbf{u}, \mathbf{v}$. For the Poincaré disk model, when the points $\mathbf{x}, \mathbf{y}$ are near the boundary of the unit disk, the floating point representation of the norm values $\|\mathbf{x}\|^2, \|\mathbf{y}\|^2$ becomes one. The denominator of the geodesic distance then becomes zero. For the Lorentz model, if $\mathbf{u} = \mathbf{v}$ and $\mathbf{u}$ contains large values in the coordinates, $\alpha$ becomes less than one which results in NaN because the domain of $\cosh^{-1}(x)$ is $x \geq 1$.

## 2.2 Statistical manifold of univariate Gaussians

A particular case of the Riemannian manifold is a statistical manifold, where each point in the manifold corresponds to a probability distribution. Specifically, the parameter manifold $\mathcal{M}$ of the probability distributions $p_\theta : \mathcal{X} \to \mathbb{R}$, where $\theta \in \mathcal{M}$, equipped with the Fisher information metric (FIM) forms a Riemannian manifold (Rao, 1992). The FIM is defined as:

$$g_{ij}(\boldsymbol{\theta}) = \int_{\mathcal{X}} \frac{\partial \log p_\theta(x)}{\partial \theta_j} \frac{\partial \log p_\theta(x)}{\partial \theta_j} p_\theta(x) \, dx.$$

In the parameter space of univariate Gaussian distributions $\{(\mu, \sigma) \mid \mu \in \mathbb{R}, \sigma \in \mathbb{R}_{>0}\}$, the FIM can be simplified as two-dimensional diagonal matrix $\sigma^{-2}\text{diag}(1, 2)$ (Costa et al., 2015).

**Connection to the Poincaré half-plane model.** The diagonal form of the FIM implies that the Riemannian manifold with $\{(\mu, \sigma) \mid \mu \in \mathbb{R}, \sigma \in \mathbb{R}_{>0}\}$ has the same set of points as the manifold of the Poincaré half-plane, but with different curvature value of $-0.5$.

The parameter space of the $n$-dimensional diagonal Gaussian distributions becomes the product of $n$ manifolds of the parameter space of univariate Gaussian distributions. In turn, the statistical manifold of $n$-dimensional diagonal Gaussian distributions can be viewed as the product of $n$ hyperbolic spaces. The operations on the product of the Riemannian manifolds $\bigotimes_{i=1}^n \mathcal{M}_i$ are defined manifold-wise. For example, an exponential map applied on a point $(p_i)_{i=1}^n \in \bigotimes_{i=1}^n \mathcal{M}_i$, with tangent vector $v_i \in \mathcal{T}_{p_i}\mathcal{M}_i$ for each $i \in \{1, \cdots, n\}$, can be represented as $(\exp_{p_i}(v_i))_{i=1}^n$.

**Distance in the statistical manifold.** In the statistical manifold, distance functions measure the difference between two distributions on the statistical manifold. One example is the geodesic distance derived from the FIM, which is called the Fisher-Rao distance. The Fisher-Rao distance of the statistical manifold of univariate Gaussian distributions is the same as the geodesic distance of the Poincaré half-plane model with constant negative curvature $-0.5$.

KL divergence is another widely-used statistical distance for distributions, defined as $D_{\text{KL}}(p \parallel q) := \int_x p(x) \log \frac{p(x)}{q(x)} \, dx$ for two distributions $p, q$ in the same statistical manifold. One notable property of KL divergence is that it can locally approximate the squared Fisher-Rao distance (Tifrea et al., 2018):

$$D_{\text{KL}}(p(\cdot; \boldsymbol{\theta} + d\boldsymbol{\theta}) \parallel p(\cdot; \boldsymbol{\theta})) = \frac{1}{2}\sum_{ij} g_{ij}(\boldsymbol{\theta})d\theta_i d\theta_j + \mathcal{O}(\|d\boldsymbol{\theta}\|^3).$$

## 3 Method

In this section, we first present the concept of the Gaussian manifold, which can have an arbitrary curvature by reparameterizing univariate Gaussian distribution. We then propose a pseudo Gaussian manifold normal distribution. Finally, we suggest a new variant of the VAE defined over the Gaussian manifold with PGM normal as prior. We denote the density function of the Gaussian distribution as $\mathcal{N}(x; \mu, \sigma^2) = 1/(\sqrt{2\pi\sigma^2}) \exp\left(-(\mu - x)^2/(2\sigma^2)\right)$.

## 3.1 Gaussian manifold with arbitrary curvature

Previous studies on hyperbolic space emphasize the importance of having an arbitrary curvature (Skopek et al., 2019; Mathieu et al., 2019). These works empirically show that the generalization performances of hyperbolic VAEs can be improved with varying curvatures. However, as shown in Section 2.2, the univariate Gaussian distributions form a manifold with curvature value of $-0.5$, limiting the flexibility of the manifold.

We show that the statistical manifold of univariate Gaussian distributions can have an arbitrary curvature by reparameterizing the univariate Gaussian distribution properly. Let $\mathcal{N}(\sqrt{2c}\mu, \sigma^2)$ be the reparameterized univariate Gaussian distribution with additional parameter $c > 0$. The reparameterization leads to the FIM of $\sigma^{-2}\mathrm{diag}(1, 1/c)$ showing that the curvature of the statistical manifold is $-c$. The computation of the sectional curvature of the extended FIM is described in Appendix B.1.

We call the statistical manifold with the reparameterized univariate Gaussian distributions and the extended FIM as the Gaussian manifold and denote it as $\mathcal{G}_c$, where $-c$ is the curvature of the Gaussian manifold.

We then verify that the KL divergence between the distributions of the Gaussian manifold approximates the geodesic distance, even in the presence of arbitrary curvature in the Gaussian manifold. Let $(\mu, \sigma) \in \mathcal{G}_c$ be an arbitrary point of the Gaussian manifold. The KL divergence between $(\mu, \sigma)$ and its neighbor $(\mu + d\mu, \sigma + d\sigma)$ can be computed as:

$$\frac{D_{\mathrm{KL}}(\mathcal{N}(\sqrt{2c}(\mu + d\mu), (\sigma + d\sigma)^2)||\mathcal{N}(\sqrt{2c}\mu, \sigma^2))}{2c} = \frac{1}{2}\begin{pmatrix} d\mu \\ d\sigma \end{pmatrix}^{\top} \begin{pmatrix} \frac{1}{\sigma^2} & 0 \\ 0 & \frac{1}{c\sigma^2} \end{pmatrix} \begin{pmatrix} d\mu \\ d\sigma \end{pmatrix} + \mathcal{O}((d\sigma)^3), \tag{1}$$

where the first term is the squared Riemannian norm of the tangent vector $(d\mu, d\sigma)$ approximating the squared Fisher-Rao distance. The detailed derivation of the KL divergence of the Gaussian manifold is described in Appendix B.2.

## 3.2 Pseudo Gaussian manifold normal distribution

We propose a pseudo Gaussian manifold (PGM) normal distribution defined over the Gaussian manifold. Let $(\mu, \sigma) \in \mathcal{G}_c$ be a point in the Gaussian manifold. Inspired by the Riemannian normal distribution (Pennec, 2006), we define the probability density function of PGM normal with the KL divergence as:

$$\mathcal{K}_c(\mu, \sigma; \alpha, \beta, \gamma) = \frac{\sigma^3}{Z(c, \beta, \gamma)} \times \exp\left(-\frac{D_{\mathrm{KL}}(\mathcal{N}(\sqrt{2c}\cdot\mu, \sigma^2) \| \mathcal{N}(\sqrt{2c}\cdot\alpha, \beta^2))}{2c \cdot \gamma^2}\right), \tag{2}$$

where $(\alpha, \beta) \in \mathcal{G}_c$, and $\gamma \in \mathbb{R}_{>0}$ are the parameters of the distribution. The distribution is centered at $(\alpha, \beta)$ with additional scale parameter $\gamma$. We verify the convergence of the PGM normal and compute the normalizing constant $Z(c, \beta, \gamma)$ over the probability measure of the Gaussian manifold at Appendix C.1. As shown in Equation 1, the KL divergence of the Gaussian manifold approximates the Fisher-Rao distance between $\mathcal{N}(\sqrt{2c}\cdot\alpha, \beta^2)$ and $\mathcal{N}(\sqrt{2c}\cdot\mu, \sigma^2)$. Therefore, the PGM normal accounts for the geometric structure of the univariate Gaussian distributions.

The factorization of the probability density function in Equation 2 multiplied with the square root of the determinant of the FIM shows the advantages of the PGM normal, which can be written as:

$$\mathcal{K}_c(\mu, \sigma; \alpha, \beta, \gamma) \cdot \sqrt{\det(g)} = \mathcal{N}(\mu; \alpha, \beta^2\gamma^2) \cdot 2\sigma\,\mathrm{Gamma}\left(\sigma^2; \frac{1}{4c\gamma^2} + 1, \frac{1}{4c\beta^2\gamma^2}\right), \tag{3}$$

where $\mathrm{Gamma}(z; a, b) = \frac{b^a}{\Gamma(a)}z^{a-1}\exp(-bz)$ and $g$ is the FIM of the Gaussian manifold. The $\sqrt{\det g}$ term enables us to sample and compute the KL divergence in an Euclidean manner. Thanks to the properties of Gaussian and gamma distributions, the PGM normal is easy to sample and has a closed-form KL divergence. The detailed derivation is available in Appendix C.2 and Appendix C.3. The factorization has the same form as the well-known conjugate prior to the Gaussian distribution. In that sense, the PGM normal explicitly incorporates the geometric structure between Gaussians into the known prior distribution.

We note that the PGM normal can be easily extended for the diagonal Gaussian manifold, a manifold formed by diagonal Gaussian distributions since the diagonal Gaussian manifold is the product of the Gaussian manifolds.

### 3.3 Gaussian manifold VAE

We introduce a Gaussian manifold VAE (GM-VAE) whose latent space is defined over the diagonal Gaussian manifold with the help of the PGM normal. We use the PGM normal for variational and prior distributions. To be specific, with the PGM normal, the evidence lower bound (ELBO) of the GM-VAE can be formalized with the diagonal Gaussian manifold $\{(\boldsymbol{\mu}, \Sigma) \mid \boldsymbol{\mu} \in \mathbb{R}^n, \Sigma \in \mathbb{R}^n_{>0}\}$ as:

$$\mathbb{E}_{q_\phi(\boldsymbol{\mu}, \Sigma \mid \mathbf{x}) \cdot \sqrt{\det(g)}} \left[ \log p_\theta(\mathbf{x} \mid \boldsymbol{\mu}, \Sigma) \right] - D_{\mathrm{KL}} \left( q_\phi(\boldsymbol{\mu}, \Sigma \mid \mathbf{x}) \cdot \sqrt{\det(g)} \parallel p(\boldsymbol{\mu}, \Sigma) \cdot \sqrt{\det(g)} \right),$$

where $p_\theta(\mathbf{x} \mid \boldsymbol{\mu}, \Sigma)$ is the decoder network, $q_\phi(\boldsymbol{\mu}, \Sigma \mid \mathbf{x})$ is the encoder network and $p(\boldsymbol{\mu}, \Sigma)$ is the prior. The variational distribution is set to $q_\phi(\boldsymbol{\mu}, \Sigma \mid \mathbf{x}) = \mathcal{K}_c(\alpha_\phi(\mathbf{x}), \beta_\phi(\mathbf{x}), \gamma_\phi(\mathbf{x}))$, where $\alpha_\theta(\mathbf{x}) \in \mathbb{R}^n$ and $\beta_\phi(\mathbf{x}), \gamma_\phi(\mathbf{x}) \in \mathbb{R}^n_{>0}$, and the prior is set to $p(\boldsymbol{\mu}, \Sigma) = \mathcal{K}_c(\mathbf{0}, I, I)$ in our experiments given curvature $-c$. The pseudo-algorithm for the decoder of GM-VAE is present at Algorithm 1.

## 4 Related Work

**Information geometry on VAE.** Focusing on the bridge between probability theory and differential geometry, the adaptation of information geometry to the deep learning framework has been investigated in various aspects (Karakida et al., 2019; Bay & Sengupta, 2017; Gomes et al., 2022). Having said that, Han et al. (2020) show that the training process of VAE can be seen as minimizing the distance between the two statistical manifolds: manifolds with the parameters of the decoder and the encoder. Not only can the parameters but the outputs from the VAE decoder be modeled as probability distributions. Arvanitidis et al. (2021) suggest a method of using the pull-back metric defined with arbitrary decoders on the latent space. Our work focuses more on the statistical manifolds lying on the outputs of the encoder with the benefits from the information geometry.

**VAE with Riemannian manifold latent space.** The latent space of VAE reflects the geometrical property of the representations of the data. The efficacy of setting the latent space to be hyperbolic space (Mathieu et al., 2019; Nagano et al., 2019; Cho et al., 2022) or elliptic space (Xu & Durrett, 2018; Davidson et al., 2018) has been verified for various datasets. Skopek et al. (2019) further extend the approach to enable the latent space to be the product of Riemannian manifolds with different learnable curvatures. On top of these, we explore the method of setting the latent space to be the diagonal Gaussian manifold, which can be viewed as the product of hyperbolic spaces, and provide a novel viewpoint on prior work with information geometry.

**Distributions over hyperbolic space.** Defining a tractable distribution over hyperbolic space is challenging. Nagano et al. (2019) suggest hyperbolic wrapped normal distribution (HWN) from the observation that the tangent space is Euclidean space. Leveraging operations defined on the tangent spaces, e.g., parallel transport, enables an easy sampling algorithm. Mathieu et al. (2019) propose a rejection sampling method for the Riemannian normal distribution defined on the Poincaré disk model, namely Poincaré normal distribution. This method rejects the pathological samples and enables accurate sampling from the distribution in exchange for high computational complexity.

Although these distributions are widely adopted in many applications (Skopek et al., 2019; Mathieu & Nickel, 2020; Cho et al., 2022), one can barely adopt the full covariance matrix due to the difficulties

---

**Algorithm 1** Decoder

---

**Input** Parameter $(\alpha, \beta) \in \mathcal{G}_c$, $\gamma$, Decoding layers $\mathrm{Dec}(\cdot)$
**Output** Reconstruction $\mathbf{x}'$

1: Sample $\mu \sim \mathcal{N}(\alpha, \beta^2 \gamma^2)$
2: Sample $\log \sigma^2 \sim \mathrm{Gamma}\left(\frac{1}{4c\gamma^2} + 1, \frac{1}{4c\beta^2\gamma^2}\right)$
3: $\mathbf{x}' = \mathrm{Dec}([\mu, \log \sigma^2])$
4: **return** $\mathbf{x}'$

---

Table 1: Density estimation on real-world datasets. $d$ denotes the latent dimension. We report the negative test log-likelihoods of average 10 runs for Breakout, CUB, Food101, and Oxford102 with 95% confidence interval. N/A in the log-likelihood indicates that the results are not available due to the failure of all runs, and N/A in the standard deviation indicates the results are not available due to failures of some runs. The best results are bolded.

| | $d$ | $\mathcal{E}$-VAE | $\mathcal{L}$-VAE | $\mathcal{P}$-VAE | GM-VAE $(c = 1)$ | GM-VAE $(c = 1/2)$ | GM-VAE $(c = 3/2)$ |
|---|---|---|---|---|---|---|---|
| Breakout | 2 | $124.74_{\pm0.86}$ | $122.58_{\text{N/A}}$ | $270.05_{\pm2.84}$ | $\mathbf{121.52_{\pm1.00}}$ | $122.64_{\pm1.13}$ | $122.47_{\pm1.98}$ |
| | 4 | $66.39_{\pm0.76}$ | $66.70_{\pm0.32}$ | $271.73_{\pm42.95}$ | $65.83_{\pm0.49}$ | $66.39_{\pm0.50}$ | $\mathbf{65.80_{\pm0.49}}$ |
| | 8 | $\mathbf{44.97_{\pm0.37}}$ | $45.25_{\pm0.27}$ | $81.55_{\pm64.61}$ | $45.14_{\pm0.30}$ | $45.31_{\pm0.36}$ | $45.36_{\pm0.49}$ |
| CUB | 50 | $992.05_{\pm1.38}$ | $993.03_{\pm1.64}$ | $990.49_{\pm2.26}$ | $985.46_{\pm3.82}$ | $986.27_{\pm3.81}$ | $\mathbf{979.14_{\pm3.70}}$ |
| | 60 | $969.99_{\pm3.13}$ | $968.79_{\pm3.70}$ | $964.02_{\pm3.55}$ | $958.00_{\pm3.25}$ | $960.88_{\pm3.46}$ | $\mathbf{956.77_{\pm2.53}}$ |
| | 70 | $949.13_{\pm2.72}$ | $948.88_{\pm3.19}$ | $944.24_{\pm4.40}$ | $939.08_{\pm3.12}$ | $942.34_{\pm3.44}$ | $\mathbf{937.15_{\pm2.76}}$ |
| Food101 | 50 | $1297.81_{\pm4.51}$ | $1298.45_{\pm6.32}$ | $1293.26_{\pm7.14}$ | $\mathbf{1286.30_{\pm6.19}}$ | $1299.58_{\pm7.02}$ | $1290.57_{\pm8.23}$ |
| | 60 | $1224.03_{\pm8.31}$ | $1227.16_{\pm5.18}$ | $1218.09_{\pm3.88}$ | $1213.31_{\pm3.88}$ | $1216.63_{\pm4.56}$ | $\mathbf{1207.30_{\pm5.12}}$ |
| | 70 | $1164.95_{\pm3.80}$ | $1165.39_{\pm5.54}$ | $1165.91_{\pm4.91}$ | $1152.80_{\pm3.35}$ | $1160.97_{\pm4.18}$ | $\mathbf{1149.56_{\pm3.41}}$ |
| Oxford102 | 50 | $1297.41_{\pm2.69}$ | $1296.41_{\pm1.56}$ | $1294.12_{\pm1.80}$ | $1292.90_{\pm3.43}$ | $\mathbf{1289.43_{\pm2.46}}$ | $1289.99_{\pm1.72}$ |
| | 60 | $1253.80_{\pm2.57}$ | $1256.52_{\pm2.99}$ | $1251.77_{\pm1.82}$ | $\mathbf{1245.49_{\pm2.18}}$ | $1248.72_{\pm1.62}$ | $1247.47_{\pm2.51}$ |
| | 70 | $1231.52_{\pm3.18}$ | $1229.38_{\pm3.44}$ | $1219.75_{\pm1.72}$ | $1215.07_{\pm2.52}$ | $1218.54_{\pm3.85}$ | $\mathbf{1214.85_{\pm2.56}}$ |

in Monte-Carlo based KL approximation. The number of samples to approximate the KL divergence increases exponentially when the full covariance matrix is used (Cho et al., 2022), so it is common to use isotropic or diagonal covariance instead. Especially in the Poincaré normal, the computation of KL divergence is slow due to the expensive rejection sampling.

**RL with hyperbolic space.**    The hierarchical relationship between the states lying on the trajectories earned from RL agents has been gaining attention recently. Nagano et al. (2019) have studied that the hierarchical structure of Atari2600 Breakout game states can be well-captured with hyperbolic VAEs. We compare the same task, where GM-VAE outperforms the previous work. Cetin et al. (2022) suggest using hyperbolic space as the geometric prior for representation learning in model-free RL agent, showing improvements in generalization performances. Here, we focus on model-based RL, especially the method of using the world model (Ha & Schmidhuber, 2018; Hafner et al., 2020), and open the possibility of applying hyperbolic space to broader domains of RL by solving the bottleneck of the numerical stability.

## 5    Experiments

In this section, we demonstrate the performances of GM-VAE on two tasks: density estimation of image datasets and model-based RL. We remark on the practical properties of GM-VAE shown in the experiments with additional analyses.

### 5.1    Density estimation on image datasets

We conduct density estimation on image datasets to measure the effectiveness of hyperbolic latent space against Euclidean space with the proposed GM-VAE. We use three datasets: the images from Atari2600 Breakout with binarization (Breakout) (Nagano et al., 2019), Oxford 102 Flower (Oxford102) (Nilsback & Zisserman, 2008), Food101 (Bossard et al., 2014), and Caltech-UCSD Birds-200-2011 (CUB) (Wah et al., 2011). The datasets are chosen with the four lowest $\delta$-hyperbolicity ($\delta$-H), a metric that measures how the given images are well-embed in hyperbolic space. Low $\delta$-H implies that the dataset is likely to embed in hyperbolic space. The details about $\delta$-H are available in Appendix D. The values of $\delta$-H for the four datasets and other candidate datasets are in Appendix D.2. Several studies show that the images from the chosen datasets have an implicit hierarchical structure (Nagano et al., 2019; Li et al., 2019; Bossard et al., 2014; Kerdels & Peters, 2015).

We compare GM-VAE with the three baseline models: VAE with Euclidean latent space ($\mathcal{E}$-VAE), and hyperbolic VAE equipped with HWN ($\mathcal{L}$-VAE) and Poincaré normal ($\mathcal{P}$-VAE). We use the product

latent space for both $\mathcal{L}$-VAE and $\mathcal{P}$-VAE, and set the curvature value to $-1$. The other details on the implementation and experimental setups are described in Appendix E.1.

The results are reported at Table 1. GM-VAE outperforms the baselines in all the settings, except one case of Breakout. Especially in CUB and Oxford102, GM-VAE outperforms the baselines regardless of the curvature value. In Breakout, $\mathcal{P}$-VAE shows inferior performance due to unstable training, and $\mathcal{L}$-VAE fails in some of the runs with small latent dimension. The results of $\mathcal{P}$-VAE and $\mathcal{L}$-VAE with non-product latent space, a common choice in previous work, are also present in Appendix F.

## 5.2 State representation learning in model-based RL

We focus on the model-based RL task to verify the utility of GM-VAE on various tasks. Specifically, we apply GM-VAE to a world model, which aims to learn the representation of the environments (Ha & Schmidhuber, 2018; Hafner et al., 2019a,b, 2020). We use DreamerV2 (Hafner et al., 2020) as the baseline model to evaluate the performance of GM-VAE in modeling environments. DreamerV2 is composed of a recurrent state space model (RSSM) (Hafner et al., 2019b) and three predictors for the image $p_\phi(x_t|h_t, z_t)$, the reward $p_\phi(r_t|h_t, z_t)$, and the discount factor $p_\phi(\gamma_t|h_t, z_t)$, where $x_t$ is the observation which the format is the image, $r_t$ is the reward, $\gamma_t$ is the discounting factor, $h_t$ is the deterministic recurrent state, and $z_t$ is the stochastic state. The model is trained by maximizing the likelihood of $p(\mathbf{x}, \mathbf{r}, \boldsymbol{\gamma} \mid \mathbf{a})$ given observations $\mathbf{x}$, rewards $\mathbf{r}$, and discount factors $\boldsymbol{\gamma}$ earned from the sequence of actions $\mathbf{a}$ of an agent. By deriving the evidence lower bound of $p(\mathbf{x}, \mathbf{r}, \boldsymbol{\gamma} \mid \mathbf{a})$, the world model is learned to optimize the likelihood with the variational distribution $q_\theta(z_t \mid h_t, x_t)$ with the following objective $\mathcal{L}(\phi, \theta)$ as:

$$\mathcal{L}(\phi, \theta) = \mathbb{E}\left[ \sum_{t=1}^{T} \Big( -\log p_\phi(x_t, r_t, \gamma_t|h_t, z_t) + \beta \, \mathrm{D_{KL}}\left[ q_\theta(z_t|h_t, x_t) \parallel p_\phi(z_t|h_t) \right] \Big) \right],$$

where $\beta$ is KL loss scaling factor, $T$ is the length of input sequence, $p_\phi$ is the prior, and $q_\phi$ is the approximated posterior. GM-VAE is employed by replacing the space of $z_t$ with the Gaussian manifold and two components in RSSM, the representation model $q_\theta(z_t|h_t, x_t)$ and transition predictor $p_\phi(z_t|h_t)$, with PGM normal.

We compare evaluation scores between different types of latent space on world model learning over the Atari2600 environments. The agents are trained with 100M environment steps. We select games having the $\delta$-H values of the four lowest and the two highest among 60 popular Atari2600 games. The other details on the implementation and experimental setups are described in Appendix E.2 with the $\delta$-H for all 60 games in Appendix D.3.

With a commonly-used hyperbolic distribution, i.e., HWN, we observe that training the world model fails due to the numerical stability issue. On the other hand, GM-VAE shows competitive results with the baselines in Euclidean and discrete latent space in all the games we test. The results are reported in Figure 2b. We note that the reproduced Euclidean baseline results by using the official code are better than those reported in Hafner et al. (2020).

## 5.3 Remark on GM-VAE

**Numerical stability.** One notable property of GM-VAE is the numerical stability during training compared to $\mathcal{L}$-VAE and $\mathcal{P}$-VAE. During the experiments, $\mathcal{L}$-VAE and $\mathcal{P}$-VAE fail to run in some of the Breakout image density estimations and all the seeds of model-based RL due to the numerical instability. Similar observations are also reported in several previous works (Mathieu et al., 2019; Chen et al., 2021; Skopek et al., 2019). The sampling from a hyperbolic distribution is a major cause of the numerical instability. Consequently, the sampling-based KL divergence computation can be unstable.

We first show that sampling from PGM normal can be stabilized via a simple reparameterization trick. To train GM-VAE, one needs to obtain sample $\mu$ and $\sigma$ from PGN normal $\mathcal{K}_c(\alpha, \beta, \gamma)$. Sampling $\mu$ can be done from $\mathcal{N}(\alpha, \beta^2\gamma^2)$ without numerical issues. Sampling $\sigma$ can be done from $\mathrm{Gamma}(a, b)$ where $a = 1/4c\gamma^2 + 1, b = 1/4c\beta^2\gamma^2$ as shown in Appendix C.2. However, due to machine precision error, often, $\beta$ violates the manifold constraints, i.e., $\beta = 0$. Eventually, direct sampling of $\sigma$ can cause the numerical instability. To avoid $\beta$ being zero, we use the output of the VAE encoder as $\log \beta^2$ whose value ranges over the entire real numbers and is more stable even when $\beta$ is close to

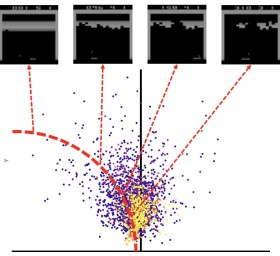

| Latent space | Euc. | Disc. | Hyp. | $\delta-$H |
|---|---|---|---|---|
| Breakout | **329.0** | 256.8 | 319.3 | 0.12 |
| Alien | 3412.5 | 3120.0 | **3485.0** | 0.14 |
| Zaxxon | 34275 | 38825 | **38950** | 0.14 |
| Ice Hockey | **25.50** | 11.80 | 20.75 | 0.14 |
| Freeway | 32.8 | **33.0** | **33.0** | 0.38 |
| Krull | 53290 | 36135 | **66185** | 0.38 |

(a) Latent space analysis        (b) Results

Figure 2: The results of model-based RL experiment. (a) The dots from yellow to purple represent the latent states from the world model in the Atari2600 Breakout with decreasing rewards. Along the red geodesic dashed line passing, we sample for images to visualize the learned representations. As the sample shows, we can observe a hierarchical structure at different stages of the game along the geodesic. (b) We compare the methods of using Euclidean, discrete, and hyperbolic latent space. We report averaged rewards over four runs and bold the best reward.

zero. With $\log \beta^2$, instead of sampling $\sigma^2$, we sample $\log \sigma^2 = \log \epsilon + \log b$, where $\epsilon$ is sampled from Gamma$(a, 1)$, through the reparameterization of the Gamma distribution, where $\log b$ can be directly computed from $\log \beta^2$.

We can show that the KL divergence between an arbitrary PGM normal and prior distribution $\mathcal{K}_c(\mathbf{0}, I, I)$ has a closed-form solution without any sampling. The KL divergence of PGM normal is the sum of the KL divergences between two Gaussian distributions and between two Gamma distributions, as shown in Appendix C.3. First, the KL divergence between a univariate Gaussian distribution $\mathcal{N}(\mu, \sigma^2)$ and the prior distribution can be obtained with $\log \sigma^2$ as shown in Equation 4. Second, the KL divergence between the two Gamma distributions, Gamma$(a_1, b_1)$ and Gamma$(a_2, b_2)$, written as:

$$D_{\mathrm{KL}}(\mathrm{Gamma}(a_1, b_1) \parallel \mathrm{Gamma}(a_2, b_2)) = a_2 \log \frac{b_1}{b_2} - \ln \frac{\Gamma(a_1)}{\Gamma(a_2)} + (a_1 - a_2)\psi(a_1) - \left(1 - \frac{b_2}{b_1}\right) a_1,$$

where $\psi$ is the digamma function, can be computed using $\log b$.

**Training time comparison.** Common bottlenecks of the mode hyperbolic VAEs arise from the complex manifold constraints and the difficulty of sampling from the hyperbolic distributions. For example, the Poincaré disk model of $\mathcal{P}$-VAE and the Lorentz model of $\mathcal{L}$-VAE requires the samples to be inside of a unit disk and to be on a hyperboloid with constraint $\{\mathbf{x} \in \mathbb{R}^{n+1} \mid -x_0^2 + \sum_{i=1}^n x_i^2\}$, respectively. Such manifolds need complex transformations, e.g., clipping, projection, or geometric transformations using the Riemannian operations, to

Table 2: The training time of the VAEs in density estimation of the Breakout image dataset. We report the training time of the VAEs in seconds per epoch. GM-VAE is 1.93x faster than $\mathcal{P}$-VAE and 1.41x faster than $\mathcal{L}$-VAE in the experiments held on a single A100 40GB PCI GPU.

| $\mathcal{E}$-VAE | $\mathcal{L}$-VAE | $\mathcal{P}$-VAE | GM-VAE |
|---|---|---|---|
| 24.5 | 35.9 | 49.2 | 25.5 |

match the manifold constraint so making the training of the hyperbolic VAEs slower. The Gaussian manifold, on the other hand, has a much simple manifold constraint and even does not require any transformations if we utilize the log space of $\sigma$.

We report the time consumptions of the VAEs with the latent dimension of 8 per epoch in the density estimation of Breakout at Table 2. The results demonstrate that the algorithmic distinctions enable GM-VAE to be trained much faster than the baseline hyperbolic VAEs and even similar to $\mathcal{E}$-VAE.

**Latent space analysis.** We present a plot of the learned representation in the hyperbolic space at Figure 2a for qualitative analysis. We take the world model with GM-VAE trained for Breakout and illustrate the geodesic starting from the origin in the figure with four generated samples along with the geodesic. We also provide the scatter plot of game states with their cumulative rewards represented in different colors. The brighter the color, the higher the cumulative reward. The scatter

plot reveals that the states with high cumulative rewards are distributed near the origin. Together with the samples from the geodesic, we can observe that the hierarchical structure of Breakout is well captured in the latent space.

Note that in the Poincaré disk model, the depth of the hierarchy is expected to be shown as the distance from the origin (Nickel & Kiela, 2017). When the root node is placed near the origin, the leaf nodes are likely to be placed near the boundary of the open disk. The geodesic lines starting from the origin to the boundary of the Poincaré disk model are identical to the geodesics of the Gaussian manifold starting from $(0, 1)$, i.e., the origin of the Gaussian manifold. The connection implies that the data hierarchy should be aligned along the geodesic curves if the hierarchy is well captured.

To quantitatively measure the correlation between the cumulative rewards and the states, we measure the Pearson correlation between the cumulative reward and the norm of the states. We obtain a correlation coefficient of 0.46 from the hyperbolic latent space, whereas the correlation coefficient of the Euclidean latent space is 0.40, showing the hyperbolic space better captures the hierarchy along the increasing norm. More experimental details are explained in Appendix G.2.

## 6   Conclusion & Future Work

In this work, we propose a novel method of representation learning with GM-VAE, utilizing the Gaussian manifold for the latent space. With the newly-proposed PGM normal defined over the Gaussian manifold, which shows better stability and ease of sampling compared to the commonly-used ones, we verify the efficacy of our method on several tasks. Our method achieves outperforming results on density estimation with image datasets and competitive results on model-based RL compared to the baselines. We explain the behavior of GM-VAE in terms of solving the frequent numerical issue of commonly-used hyperbolic VAEs. The analysis of latent space exhibits that the hierarchy lying in the dataset can be preserved by using GM-VAE.

We suggest that the numerical stability of our method can be helpful for scaling the generative models, e.g., very deep VAE (Child, 2021), endowed with hyperbolic geometrical priors. As GM-VAE is beneficial for capturing hierarchy with promising results in modeling RL environment, another potential future work can be extending the use of hyperbolic space, such as learning a skill tree for solving complex long-horizon tasks (Shi et al., 2022). We believe that the connection between the statistical manifold and hyperbolic space provides new insight to the research community and hope to see more interesting connections and analyses in the future.

## Acknowledgments and Disclosure of Funding

This work was partly supported by Institute of Information & communications Technology Planning & Evaluation (IITP) grant funded by the Korea government (MSIT) (No.2019-0-01906, Artificial Intelligence Graduate School Program (POSTECH)) and National Research Foundation of Korea (NRF) grant funded by the Korea government (MSIT) (No. RS-2023-00217286) and National Research Foundation of Korea (NRF) grant funded by the Korea government (MSIT) (NRF-2021R1C1C1011375)

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

# A Machine Precision Error Analysis on Hyperbolic Space

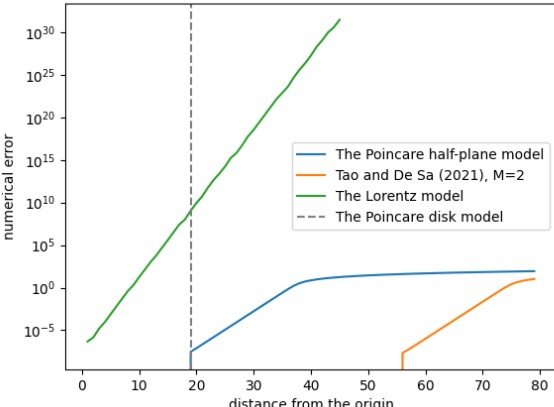

Figure 3: The machine precision errors of the hyperbolic models. For the Poincare half-plane model, we report the upper bound of the machine precision error derived by Yu & Sa (2021). For the Lorentz model, we report the error between the Lorentzian product of the point and -1, which is the manifold constraint of the Lorentz model. For the Poincare disk model, we report the threshold that the norm of the point becomes one. The precision is set to 32 bits. The analysis reveals that all the three models suffer from numerical stability issue with points which the distance between the origin is farther than 20.

# B Gaussian Manifold

## B.1 Curvature of the Gaussian manifold

We construct a Riemannian manifold $\{(\mu, \sigma) \mid \mu \in \mathbb{R}, \sigma \in \mathbb{R}_{>0}\}$ with a positive constant $c$ and the metric tensor $\sigma^{-2}\mathrm{diag}(1, 1/c)$, which we will name Gaussian manifold. We need to show the value of the curvature.

First, we need to compute the Christoeffel symbols of the Gaussian manifold defined as:

$$\Gamma_{ij}^k = \frac{1}{2}g^{kl}\left(\frac{\partial g_{jl}}{\partial g_i} + \frac{\partial g_{il}}{\partial g_j} - \frac{\partial g_{ij}}{\partial g_l}\right),$$

where $g_{ij}$ is the $(i,j)$ element of the metric tensor and $g^{ij}$ is the $(i,j)$ element of the inverse of the metric tensor.

The Christoeffel symbols of the Gaussian manifold are:

$$\Gamma_{ij}^1 = \begin{pmatrix} 0 & -\frac{1}{\sigma} \\ -\frac{1}{\sigma} & 0 \end{pmatrix}$$

$$\Gamma_{ij}^2 = \begin{pmatrix} \frac{c}{\sigma} & 0 \\ 0 & -\frac{1}{\sigma} \end{pmatrix}.$$

Then, the sectional curvature of the space $\kappa_g$ with given tangent vectors $du, dv$ is computed as:

$$\begin{aligned}
\kappa_g &= \frac{\langle R(d\mu, d\sigma)d\sigma, d\mu\rangle}{\det g} \\
&= \frac{1}{\det g} \cdot g_{1m}\left(\frac{\partial \Gamma_{22}^m}{\partial \mu} - \frac{\partial \Gamma_{12}^m}{\partial \sigma} + \Gamma_{22}^p\Gamma_{1p}^m - \Gamma_{12}^p\Gamma_{2p}^m\right) \\
&= \frac{-\frac{1}{\sigma^4}}{\frac{1}{c\sigma^4}} \\
&= -c.
\end{aligned}$$

Note that the sectional curvature of two-dimensional Riemannian manifold is same as the Gaussian curvature where $\langle d\mu, d\mu \rangle \langle d\sigma, d\sigma \rangle - \langle d\mu, d\sigma \rangle^2 = \det g$.

## B.2   Gaussian manifold with KL-divergence

Between two univariate Gaussian distributions $\mathcal{N}(\mu_1, \sigma_1^2)$ and $\mathcal{N}(\mu_2, \sigma_2^2)$, we can compute the KL divergence as:

$$D_{\mathrm{KL}}(\mathcal{N}(\mu_1, \sigma_1^2) \parallel \mathcal{N}(\mu_2, \sigma_2^2)) = \frac{1}{2}\left( \log \frac{\sigma_2^2}{\sigma_1^2} + \frac{\sigma_1^2 + (\mu_1 - \mu_2)^2}{\sigma_2^2} - 1 \right). \tag{4}$$

We extend the KL divergence for an arbitrary curvature of the Gaussian manifold as:

$$D_{\mathrm{KL}}^{\mathcal{G}_c}((\mu_1, \sigma_1), (\mu_2, \sigma_2)) := \frac{D_{\mathrm{KL}}(\mathcal{N}(\sqrt{2c}\mu_1, \sigma_1^2) \parallel \mathcal{N}(\sqrt{2c}\mu_2, \sigma_2^2))}{2c}.$$

Now, we show that the extended KL divergence still approximates the Riemannian distance of the manifold as:

$$
\begin{aligned}
D_{\mathrm{KL}}^{\mathcal{G}_c}((\mu + d\mu, \sigma + d\sigma), (\mu, \sigma)) &= \frac{1}{2 \cdot 2c}\left( \log \frac{\sigma^2}{(\sigma + d\sigma)^2} + \frac{(\sigma + d\sigma)^2 + 2c(d\mu)^2}{\sigma^2} - 1 \right) \\
&= \frac{1}{2 \cdot 2c}\left( -2\log\left(1 + \frac{d\sigma}{\sigma}\right) + \frac{2\sigma d\sigma + (d\sigma)^2}{\sigma^2} + \frac{2c(d\mu)^2}{\sigma^2} \right) \\
&= \frac{1}{2 \cdot 2c}\left( -2\left(\frac{d\sigma}{\sigma} - \frac{(d\sigma)^2}{2\sigma^2}\right) + \frac{2\sigma d\sigma + (d\sigma)^2}{\sigma^2} + \frac{2c(d\mu)^2}{\sigma^2} + \mathcal{O}((d\sigma)^3) \right) \\
&= \frac{1}{2}\begin{pmatrix} d\mu \\ d\sigma \end{pmatrix}^T \begin{pmatrix} \frac{1}{\sigma} & 0 \\ 0 & \frac{1}{c\sigma^2} \end{pmatrix}\begin{pmatrix} d\mu \\ d\sigma \end{pmatrix} + \mathcal{O}((d\sigma)^3).
\end{aligned}
$$

## C   Pseudo Gaussian Manifold Normal Distribution

In this section, we derive the normalizing constant $Z(c, \beta, \gamma)$ and the factorization of the PGM normal which the density function is defined as:

$$\mathcal{K}_c(\mu, \sigma; \alpha, \beta, \gamma) = \frac{\sigma^3}{Z(c, \beta, \gamma)}\exp\left( -\frac{D_{\mathrm{KL}}^{\mathcal{G}_c}((\mu, \sigma), (\alpha, \beta))}{\gamma^2} \right).$$

### C.1   Normalizing Constant

The given probability density function needs to satisfy the following condition:

$$\int_{\mathcal{G}_c} \mathcal{K}_c(\mu, \sigma; \alpha, \beta, \gamma)\sqrt{\det g} \cdot d(\mu, \sigma) = 1, \tag{5}$$

where $\sqrt{\det g} \cdot d(\mu, \sigma)$ is the probability measure over the Gaussian manifold induced from the Lebesgue measure $d(\mu, \sigma)$. The normalizing factor $Z(c, \beta, \gamma)$ can be computed using the condition

 as:

$$Z(c, \beta, \gamma) = \int_0^\infty \int_{-\infty}^\infty \sigma^3 \cdot \exp\left(-\frac{D_{\mathrm{KL}}^{\mathcal{G}_c}((\mu, \sigma), (\alpha, \beta))}{\gamma^2}\right) \frac{1}{\sqrt{c}\sigma^2} \, d\mu \, d\sigma$$

$$= \frac{1}{\sqrt{c}} \left(\beta^{-\frac{1}{2c\gamma^2}} \exp\left(\frac{1}{4c\gamma^2}\right) \int_0^\infty \sigma \cdot (\sigma^2)^{\left(\frac{1}{4c\gamma^2}+1\right)-1} \exp\left(-\frac{\sigma^2}{4c\beta^2\gamma^2}\right) d\sigma\right)$$
$$\times \left(\int_{-\infty}^\infty \exp\left(-\frac{(\mu-\alpha)^2}{2\beta^2\gamma^2}\right) d\mu\right)$$

$$= \frac{1}{2\sqrt{c}} \sqrt{2\pi}\beta^3\gamma \exp\left(\frac{1}{4c\gamma^2}\right) \Gamma\left(\frac{1}{4c\gamma^2}\right) \left(\frac{1}{4c\gamma^2}\right)^{-\frac{1}{4c\gamma^2}}$$
$$\times \left(\int_0^\infty \mathrm{Gamma}\left(\sigma^2; \frac{1}{4c\gamma^2}+1, \frac{1}{4c\beta^2\gamma^2}\right) d\sigma^2\right) \left(\int_{-\infty}^\infty \mathcal{N}(\mu; \alpha, \beta\gamma) \, d\mu\right) \tag{6}$$

$$= \frac{\sqrt{2\pi}\beta^3}{2\sqrt{c}} \gamma \exp\left(\frac{1}{4c\gamma^2}\right) \Gamma\left(\frac{1}{4c\gamma^2}\right) \left(\frac{1}{4c\gamma^2}\right)^{-\frac{1}{4c\gamma^2}}.$$

Finally, the logarithm of the normalizing factor is computed as:

$$\log Z(c, \beta, \gamma) = \frac{1}{2}\log(2\pi) + 3\log\beta - \frac{1}{2}\log c - \log 2 + \frac{1}{2}\log\gamma^2 + \log\Gamma\left(\frac{1}{4c\gamma^2}\right) + \frac{1}{4c\gamma^2}(1+\log(4c\gamma^2)).$$

## C.2 Sampling

Suppose that $p(\mu, \sigma) = \mathcal{K}_c(\mu, \sigma; \alpha, \beta, \gamma)\sqrt{\det g}$. Sampling $\mu$ and $\sigma$ from the probability distribution $p(\mu, \sigma)$ can be done by sampling $\mu$ from the marginal distribution $p(\mu)$ and then sampling $\sigma$ from the conditional distribution $p(\sigma|\mu)$. The marginal distribution $p(\mu)$ can be derived from  as:

$$p(\mu) = \int_0^\infty p(\mu, \sigma) \, d\sigma$$
$$= \int_0^\infty \mathcal{K}_c(\mu, \sigma; \alpha, \beta, \gamma)\sqrt{\det g} \, d\sigma$$
$$= \left(\int_0^\infty \mathrm{Gamma}\left(\sigma^2; \frac{1}{4c\gamma^2}+1, \frac{1}{4c\beta^2\gamma^2}\right) d\sigma^2\right) \mathcal{N}(\mu; \alpha, \beta^2\gamma^2)$$
$$= \mathcal{N}(\mu; \alpha, \beta^2\gamma^2).$$

 also implies that $\mu$ and $\sigma$ are independent in the aspect of $p(\mu, \sigma)$ so the conditional distribution $p(\sigma|\mu)$ is identical to the marginal distribution $p(\sigma)$. The marginal distribution $p(\sigma)$ is computed as:

$$p(\sigma) = \int_{-\infty}^\infty p(\mu, \sigma) \, d\mu$$
$$= \int_{-\infty}^\infty \mathcal{K}_c(\mu, \sigma; \alpha, \beta, \gamma)\sqrt{\det g} \, d\mu$$
$$= \left(2\sigma \cdot \mathrm{Gamma}\left(\sigma^2; \frac{1}{4c\gamma^2}+1, \frac{1}{4c\beta^2\gamma^2}\right)\right) \left(\int_{-\infty}^\infty \mathcal{N}(\mu; \alpha, \beta^2\gamma^2) \, d\mu\right)$$
$$= 2\sigma \cdot \mathrm{Gamma}\left(\sigma^2; \frac{1}{4c\gamma^2}+1, \frac{1}{4c\beta^2\gamma^2}\right).$$

Here, sampling $\sigma$ from $p(\sigma)$ can be easily replaced by the procedure of sampling $\sigma^2$ from $p(\sigma^2)$, which is identical to $p(\sigma)/(2\sigma) = \mathrm{Gamma}\left(\sigma^2; \frac{1}{4c\gamma^2}+1, \frac{1}{4c\beta^2\gamma^2}\right)$, and then applying square root to the sample $\sigma^2$ to get $\sigma$.

## C.3 KL Divergence

Suppose that $p(\mu, \sigma), q(\mu, \sigma)$ are two different PGM normal multplied with $\sqrt{\det g}$. As shown in Appendix C.2, $\mu$ and $\sigma$ are independent so the KL divergence between $p, q$ is same as $D_{\mathrm{KL}}(p(\mu)\|q(\mu)) + D_{\mathrm{KL}}(p(\sigma)\|q(\sigma))$. The first term is the KL divergence between two normal distribution. The second term is same with $D_{\mathrm{KL}}(p(\sigma^2)\|q(\sigma^2))$ due to the change-of-variable formula, so it is the KL divergence between two gamma distribution.

# D   $\delta$-Hyperbolicity

In this section, we explain about $\delta$-hyperbolicity ($\delta$-H) and show the $\delta$-H values of the candidate datasets of our experiments.

## D.1   $\delta$-hyperbolicity

Given a metric space $(X, d)$, the metric space is said to be $\delta$-hyperbolic if, for any geodesic triangle, i.e., a triangle where each side is a geodesic curve, any point on the side of the geodesic triangle is within the distance of less than or equal to $\delta$ of the other two sides; when such $\delta$ exists, $(X, d)$ is said to be hyperbolic. To be specific, when the Gromov product between any three points $x, y, z \in X$ is given as:

$$(y, z)_x = \frac{1}{2} \left( d(x, y) + d(x, z) - d(y, z) \right),$$

if then the following inequality holds for any four points $x, y, z, w \in X$, we call the metric space is $\delta$-hyperbolic:

$$(x, z)_w \geq \min((x, y)_w, (y, z)_w) - \delta. \quad (7)$$

$\delta$-hyperbolicity ($\delta$-H) is defined to be the minimum value of $\delta$ satisfying Equation 7 and is used as a measurement quantifying how a given metric space well embeds in hyperbolic Khrulkov et al. (2020); Cetin et al. (2022). Figure 4 illustrates the concept of $\delta$-H. We note that the lower $\delta$-H the metric space has, the less deviation from the exact hyperbolic space is.

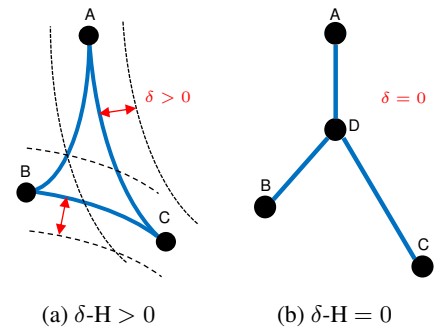

(a) $\delta$-H $> 0$      (b) $\delta$-H $= 0$

Figure 4: The illustration of $\delta$-H for given geodesic triangles. The blue lines denote the geodesic curves between the points. (a) When the side $AB$ is contained in the union of the two $\delta$-neighbor regions of the side $AC$ and $BC$, we say that the geodesic triangle is $\delta$-hyperbolic. $\delta$-H of the geodesic triangle is then defined as the minimum possible value of $\delta$. (b) Any tree-structured triangle, where the geodesic curves correspond to the tree edges, is 0-H. For example, given geodesic triangle $ABC$, one side $AB$ is already occupied by the two sides $AB$ and $BC$, as the geodesic between the points $A$ and $B$ is the union of edge $AD$ and $BD$ being occupied by other counterparts.

**Computation.** We measure the $\delta$-H values of the images $X$ from the datasets by following the procedure from Fournier et al. (2015). We first extract the embeddings of the images using a pre-trained feature extractor to construct a metric space of the images. We then randomly sample a fixed point $w$ and calculate the pairwise Gromov product of the embeddings $D$ with $w$ as Equation 7. We finally determine the $\delta$-H of the images $X$ by finding the largest coefficient of $(\max_k \min_{ij}(D_{ik}, D_{kj})) - D$.

To reduce the scale difference between the datasets, we report the value $2\delta(X)/\text{diam}(X)$ where $\text{diam}(X)$ denotes the maximum pairwise distance of $X$. Because computing the matrix $D$ among all the images $X$ is computationally expensive, we compute the $\delta$-H of randomly sampled 1,000 images from $X$. We repeat this 10 times and report the average $\delta$-H.

## D.2   Image datasets

We measure the $\delta$-H of the image datasets using an ImageNet pre-trained Inception V3 as a feature extractor.

| Breakout | CUB | Food101 | Oxford102 | CIFAR-10 | SVHN | CelebA |
|----------|-------|---------|-----------|----------|-------|--------|
| 0.124 | 0.223 | 0.233 | 0.238 | 0.248 | 0.283 | 0.287 |

## D.3   Atari2600 environments

We collect the Atari2600 images using the pre-trained agents of Gogianu et al. (2022) and measure the $\delta$-H using the image encoder from the agents. For each environment, we report the $\delta$-H of the images which are collected by the agent with the highest reward. We also report the corresponding reward. We run the agents for at least 6 episodes.

| Game | $\delta$-H | Reward | Game | $\delta$-H | Reward |
|---|---|---|---|---|---|
| Breakout | 0.12 | 340 | AirRaid | 0.25 | 11729 |
| Alien | 0.14 | 6855 | Frostbite | 0.25 | 8653 |
| Zaxxon | 0.14 | 13100 | Pooyan | 0.25 | 7973 |
| IceHockey | 0.14 | 3 | ChopperCommand | 0.26 | 10530 |
| Gravitar | 0.17 | 1004 | KungFuMaster | 0.26 | 27733 |
| Carnival | 0.17 | 5605 | YarsRevenge | 0.26 | 56405 |
| RoadRunner | 0.18 | 60050 | Phoenix | 0.27 | 30208 |
| Pong | 0.18 | 21 | JourneyEscape | 0.27 | -707 |
| Tutankham | 0.18 | 236 | SpaceInvaders | 0.27 | 15623 |
| Boxing | 0.19 | 99 | Hero | 0.28 | 28592 |
| Solaris | 0.19 | 1727 | Enduro | 0.28 | 2089 |
| WizardOfWor | 0.20 | 17217 | Assault | 0.28 | 3217 |
| Seaquest | 0.20 | 29745 | Venture | 0.29 | 1529 |
| Gopher | 0.20 | 24173 | StarGunner | 0.29 | 68417 |
| Centipede | 0.20 | 4663 | Atlantis | 0.29 | 919750 |
| PrivateEye | 0.21 | 195 | DoubleDunk | 0.30 | 18 |
| Pitfall | 0.21 | 0 | TimePilot | 0.30 | 10914 |
| ElevatorAction | 0.21 | 64917 | BeamRider | 0.30 | 7069 |
| Robotank | 0.21 | 75 | BattleZone | 0.30 | 40333 |
| MsPacman | 0.21 | 5455 | NameThisGame | 0.31 | 12925 |
| DemonAttack | 0.22 | 23876 | Skiing | 0.31 | -10021 |
| Asterix | 0.22 | 26792 | Amidar | 0.32 | 2826 |
| Qbert | 0.22 | 17364 | Asteroids | 0.34 | 1405 |
| Jamesbond | 0.22 | 886 | MontezumaRevenge | 0.34 | 0 |
| VideoPinball | 0.23 | 632563 | Bowling | 0.35 | 49 |
| UpNDown | 0.23 | 27030 | BankHeist | 0.36 | 1563 |
| FishingDerby | 0.23 | 54 | Kangaroo | 0.37 | 14283 |
| Tennis | 0.23 | 22 | CrazyClimber | 0.37 | 132517 |
| Berzerk | 0.24 | 904 | Freeway | 0.38 | 34 |
| Riverraid | 0.24 | 15103 | Krull | 0.38 | 9016 |

# E    Implementation Details

In this section, we introduce the implementation details of the experiments.

## E.1    Density estimation on image datasets

We estimate the density of the images from Atari2600 Breakout (Nagano et al., 2019), Oxford102 (Nilsback & Zisserman, 2008), and CUB (Wah et al., 2011). The images of Breakout are collected from plays with a pre-trained Deep Q-Network (Mnih et al., 2015). The size of images of all the datasets is resized to $64 \times 64$, while Breakout is binarized with a threshold value of 0.1; the threshold for Breakout is determined to visualize the components clearly.

We split the datasets into train, validation, and test. For Breakout and CUB, we split the original train set into train and validation sets. For Oxford102, because the original train set is too small, we merge the original train and test set and then split it into three splits. For Food101, we randomly sample the train set and validation set from the original train set, and also randomly sample the test set from the original test set.

| Split | Breakout | CUB | Food101 | Oxford102 |
|---|---|---|---|---|
| Train | 80,000 | 4,795 | 6,000 | 5,120 |
| Validation | 9,503 | 1,199 | 1,000 | 1,228 |
| Test | 9,934 | 5,794 | 1,000 | 1,025 |

We design the encoder and decoder similar to the generator and discriminator of DCGAN (Radford et al., 2015). The details of the architecture are at Table 3. We use learning rate 1e-3, batch size 100, and Adam optimizer for training. We use Bernoulli loss as the reconstruction loss for Breakout experiments and negative log-likelihood loss as the reconstruction loss for CUB, Food101, and Oxford102 experiments. We use the validation set for early stopping and report the negative log-likelihood on the test set with 50 importance weighted samples.

Table 3: The architectures of encoder and decoder used in the density estimation experiments. $n_c$ is the number of channels of the image, $n_d$ is the latent dimension. $n_a$ is a coefficient that depends on the VAE, i.e., $n_a$ is 2, 2, 1.5, 1.5 for $\mathcal{E}$-VAE, $\mathcal{L}$-VAE, $\mathcal{P}$-VAE, and GM-VAE, respectively.

| Encoder | | Decoder | |
|---|---|---|---|
| Layer | Size | Layer | Size |
| Input | $64 \times 64 \times n_c$ | Input | $1 \times 1 \times n_d$ |
| Convolution2D | $32 \times 32 \times 32$ | TransposedConvolution2D | $4 \times 4 \times 256$ |
| LeakyReLU | | ReLU | |
| Convolution2D | $16 \times 16 \times 64$ | TransposedConvolution2D | $8 \times 8 \times 128$ |
| LeakyReLU | | ReLU | |
| Convolution2D | $8 \times 8 \times 128$ | TransposedConvolution2D | $16 \times 16 \times 64$ |
| LeakyReLU | | ReLU | |
| Convolution2D | $4 \times 4 \times 256$ | TransposedConvolution2D | $32 \times 32 \times 32$ |
| LeakyReLU | | ReLU | |
| Linear | $n_a \cdot n_d$ | TransposedConvolution2D | $64 \times 64 \times n_c$ |

## E.2 Model-based RL

We use the official TensorFlow implementation from Dreamerv2[1] to reproduce the baseline results, i.e., with Euclidean and discrete latent space. For the hyperbolic latent space results, we apply GM-VAE by replacing the latent space of $z_t$ with the Gaussian manifold and two components in RSSM, the representation model $q_\theta(z_t|h_t, x_t)$ and transition predictor $p_\phi(z_t|h_t)$, with PGM normal. The hyperparameters are set to be the same as suggested in the original paper, except for the training environment steps being 50M for Freeway and 100M for the others as we observe converging scores.

## F  Results of Non-Product Latent Space

Previous hyperbolic VAEs are implemented with a hyperbolic space, not the product of the hyperbolic spaces. We run the non-product hyperbolic VAEs in the density estimation of image datasets. Table 4 reveals that the non-product hyperbolic VAEs fail in most of the settings and the product hyperbolic space makes the hyperbolic VAEs much more stable.

## G  Model-Based RL

### G.1  Learning curves

### G.2  Latent space analysis

We conduct an analysis of the latent space of the agents learned to play Atari2600 Breakout. The purpose of the analysis is to measure how the latent spaces well-preserve the implicit hierarchy in the trajectory of the agents. To analyze the hyperbolic latent space, we need two isometries: the isometry between the Gaussian manifold and the Poincaré disk model and the translation of the Poincaré disk model.

---

[1]https://github.com/danijar/dreamerv2

Table 4: Density estimation results of non-product hyperbolic VAEs. $d$ denotes the latent dimension. N/A in the log-likelihood indicates that the results are not available due to the failure of all runs.

|  | $d$ | $\mathcal{L}$-VAE | $\mathcal{P}$-VAE |
|---|---|---|---|
| Breakout | 2 | $124.24_{\pm 1.66}$ | $266.86_{\pm 6.01}$ |
|  | 4 | $66.20_{\pm 0.14}$ | N/A |
|  | 8 | $44.76_{\pm 0.48}$ | N/A |
| CUB | 50 | N/A | N/A |
|  | 60 | N/A | N/A |
|  | 70 | N/A | N/A |
| Oxford102 | 50 | N/A | N/A |
|  | 60 | N/A | N/A |
|  | 70 | N/A | N/A |

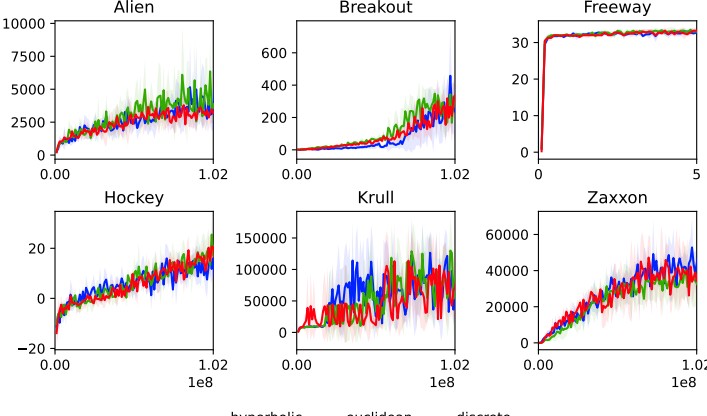

We first propose an isometry between the Gaussian manifold and the Poincaré disk model $T_{\mathcal{P}_c \to \mathcal{U}_c} : \mathcal{P}_c \to \mathcal{U}_c$:

$$T_{\mathcal{P}_c \to \mathcal{G}_c}(x, y) = \left( \frac{-2y}{(\sqrt{c}x - 1)^2 + y^2 c}, \frac{1 - (x^2 + y^2)c}{(\sqrt{c}x - 1)^2 + y^2 c} \right),$$

and the inverse is:

$$T_{\mathcal{P}_c \to \mathcal{G}_c}^{-1}(x, y) \left( \frac{\sqrt{c}x^2 + (y^2 - 1)/\sqrt{c}}{cx^2 + (y + 1)^2}, \frac{-2x}{cx^2 + (y + 1)^2} \right).$$

The translation of the Poincaré disk model can be derived using complex numbers. Let $z = x + yi \in \mathbb{C}$ and $(x, y) \in \mathcal{P}_1$ and $z_0 \in \mathbb{C}$ be the pivot point. Then the isometry that moves $z_0$ to the origin is defined as $T(z) : \mathcal{P}_1 \to \mathcal{P}_1 := \frac{z - z_0}{1 - \bar{z}_0 z}$. Note that the translation of Euclidean space is $z - z_0$.

After transforming the latents on the Gaussian manifold to the Poincaré disk model and using the translation, we can measure how the latents well-captures the hierarchical structure of data. We first pick a latent and then translate all the latents by setting the selected latent as the pivot point. We then measure the Pearson correlation between the cumulative reward of the latents and the norm.

We repeat this process for all the latents and compute the maximum of the correlations. We use the latents obtained from the agents which recorded at least 250 for long enough trajectories. We obtain a correlation coefficient of 0.46 from the hyperbolic latent space, whereas the correlation coefficient of the Euclidean latent space is 0.40, showing the hyperbolic space better captures the hierarchy along the increasing norm.

