# OpenReview forum: "Hyperbolic VAE via Latent Gaussian Distributions"
_NeurIPS.cc/2023/Conference — NeurIPS 2023 poster_

### Official Review · Reviewer_Darj · 2023-06-15

**Soundness:** 4 excellent
**Presentation:** 3 good
**Contribution:** 3 good
**Rating:** 6
**Confidence:** 4

**Summary:**

The paper proposes a pseudo-Gaussian distribution over hyperbolic spaces, which is suitable for constructing VAEs with hyperbolic latent spaces. This pseudo-Gaussian distribution arrives by replacing the usual Euclidean distance with a KL-divergence (this locally approximates the hyperbolic geodesic distance). The required properties of this distribution (sampling, normalization constant, KL-divergences) are derived and shown to be easily computable. Empirical results show promise.

**Strengths:**

1. The paper is clearly written and easy to follow.
2. The paper provides a novel (to the best of my knowledge) construction of a hyperbolic distribution that seemingly solves a practical problem (numerical instability) in hyperbolic VAEs.
3. I find it elegant how the proposed distribution factorizes into a product of known distributions (this was not obvious to me from the KL construction).

**Weaknesses:**

1. A claimed benefit of the proposed method is that it is numerically more stable than previous methods. While I do not doubt the correctness of this statement, I had wished for an empirical demonstration of the claim. As far as I can tell, numerical stability is the main practical gain over existing methods (note, this is an important gain), so I think it should be studied more extensively experimentally. In short: show me numbers that measure stability.
2. The paper mentions tuning the curvature of the hyperbolic metric, but I did not find it clear how and when such tuning takes place. Is the $c$ parameter tuned by hand?
3. Similarly, the paper mentions using a $\beta$ parameter in the VAE, but it is not clear how such a parameter is chosen in practice. Is it manually tuned? If so, does that not suggest overfitting could be a problem?
4. Since the geodesic distance is replaced by a local approximation (in the form of the KL divergence), the method is only somewhat geometric. It seems to be good enough, but I suppose that none of the established theoretical results on tree embeddings hold under the approximated metric. If so, it would be good to explicate.

*Minor comment:*
Line 126: The KL divergence is really not a "distance" and should not be described as such. Especially in a geometric paper, such confusions should be avoided.

**Questions:**

1. The KL is a local approximation to the hyperbolic metric. This is used as a distance measure in a pseudo-Gaussian distribution. I wonder how similar this construction is to a tangent space normal distribution. Both approaches rely on local approximations of the metric. Do you have comments on this link?
2. In Table 1, what does $\circ$ and $\times$ mean? I'm struggling to make sense of those symbols.


**Limitations:**

The paper does not provide much of a discussion of the limitations of the proposed method.

---

> ### Author Rebuttal · Authors · 2023-08-09
>
> We sincerely thank for providing constructive feedback. We answer the questions about i) analysis on numerical stability, ii) the parameter $c$ and $\beta$ of VAE, and iii) the approximation performance of KL divergence.
>
> ---
>
> **Q1) Quantitative analysis on the numerical stability of hyperbolic distributions.**
>
> We thank for the suggestion for strengthening our statement about numerical stability. Due to the time limit, instead of running more experiments, we conduct a machine precision error analysis on the hyperbolic distributions, which is the main cause of the numerical instability.
>
> We first analyze the causes of numerical instability of hyperbolic space with different models. The numerical issues of hyperbolic distributions occur when the mean or the sampled point does not belong to the manifold due to the machine precision error. For example, the denominator of the distance of the Poincare disk model becomes zero when the point is placed on the border of the disk. In the Lorentz model, the input of the arccosh function of the log mapping can be less than one when the mean of the distribution is far from the origin and the sampled point is near the mean point, which is out of the domain. Theoretically, these issues should not happen, but we have observed the problems during the experiments due to the limited machine precision.
>
> These operations are essential in the sampling and KL divergence computation of the distributions. Inspired by the suggestion of reviewer bHMn, we conduct a quantitative analysis on numerical stability by following [1].  We compare the machine precision errors of the three models, the Poincare half-plane model, the Lorentz model, and the Poincare disk model which are used in PGM normal, HWN, and the Poincare normal with respectively. Figure 1 shows that the Poincare half-plane model, which is used in PGM normal, is more stable than the other two models in a wider regime.
>
> Furthermore, the sampling and KL divergence computation of PGM normal are free from the manifold of the Poincare half-plane model because we can express the operations with the log-space of the half-plane manifold, which is identical to Euclidean space. For more details, see the code of density_estimation/distributions/PGMNormal/distribution.py and Appendix F of the supplementary.
>
> We also empirically show that PGM normal is the only hyperbolic distribution that runs with 16 bit precision in the RL task while the other distributions fail and output NaN.
>
> ---
>
> **Q2) The curvature $c$ of hyperbolic VAE.**
>
> The curvature of the hyperbolic space is mostly set to minus one, the best value in previous work, or learned together [6]. Some work shows that the gradient of the curvature can be computed through the Riemannian operations and carefully learning it together can enhance the performance [6]. We also conducted an experiment on learning the curvature together. Please check the “Learning curvature experiment” section of the general response for details.
>
> ---
>
> **Q3) The $\beta$ parameter of VAE.**
>
> By controlling $\beta$, we can make a tradeoff between reconstruction and KL divergence [5]. If $\beta$ is large, latents will better match with the prior, but the reconstruction performance will degrade. If $\beta$ is low, the reconstruction performance will be enhanced, but the data would be overfit at the training data. The tradeoff is an important factor in the VAE literature and frequently studied. It is usually set to one, slightly increased as the training proceeds, or follows the previous work. In the RL setting, we set $\beta$ to 0.1 which is used in the baseline model DreamerV2.
>
> ---
>
> **Q4) Tree embedding with KL divergence.**
>
> In previous work, it is empirically shown that KL divergence, the approximation of the hyperbolic distance function, can embed tree-structured data similar to the hyperbolic distance function [2, 3]. Based on the results, we believe that the approximation would not harm the capability of modeling the hierarchical structure.
>
> ---
>
> **Q5) The approximation performance between KL divergence and wrapped normal.**
>
> The geometry column of Table 1 from the main paper means that the distribution represents the density of a sample point with the hyperbolic distance. Although the HWN utilizes operations with Riemannian metric in sampling, we thought that the resulting distance is dependent on the Euclidean distance, not the hyperbolic distance due to the normal distribution on the tangent space. However, we find that the density of HWN can be seen as a local approximation of the hyperbolic distance [4], so we will remove the table in future revision.
>
> We conduct a study on the approximation performance of KL divergence and wrapped normal operation, which are used in PGM normal and HWN with respectively. The results are at Figure 4. While the KL divergence shows better approximation in points with small squared distance, the wrapped normal better approximate the squared distance in far points.
>
> ---
>
> **References**
>
> [1] Yu and De Sa, Representing Hyperbolic Space Accurately using Multi-Component Floats, NeurIPS 2021
>
> [2] Nagano et al., A Wrapped Normal Distribution on Hyperbolic Space for Gradient-Based Learning, ICML 2019
>
> [3] Cho et al., A Rotated Hyperbolic Wrapped Normal Distribution for Hierarchical Representation Learning, NeurIPS 2022
>
> [4] De Bortoli et al., Riemannian score-based generative modelling, NeurIPS 2022
>
> [5] Alemi et al., Fixing a Broken ELBO, ICML 2018
>
> [6] Skopek et al., Mixed-curvature variational autoencoders, ICLR 2020

---

> > ### Comment · Reviewer_Darj · 2023-08-15
> > **Thank you for the rebuttal**
> >
> > I will take the provided update into account when discussing the paper with the other reviewers.

---

> > > ### Author Response · Authors · 2023-08-17
> > > **Response to reviewer Darj**
> > >
> > > We appreciate responding to the rebuttal and paying attention to the contents for the further discussion. Feel free to ask any questions in the remaining discussion period!

---

### Official Review · Reviewer_bHMn · 2023-06-30

**Soundness:** 3 good
**Presentation:** 2 fair
**Contribution:** 3 good
**Rating:** 6
**Confidence:** 5

**Summary:**

This paper proposes to utilize a set of Gaussian distributions as the latent space of a VAE model. One benefit of doing so is that this forms a hyperbolic space under the Fisher information metric. A key insight is that this Gaussian manifold corresponds to the Poincare half-plane model of hyperbolic geometry. Using this construction the authors construct a Gaussian manifold VAE which locally approximates the metric and allows for fast calculation of the KL divergence and fast sampling. Empirical results demonstrate the efficacy of the proposed approach on image datasets as well as RL benchmarks.

**Strengths:**

This paper proposes an interesting way to enforce a hyperbolic latent space in a VAE setting. The connection of the FIM and Poincare half-plane using a set of univariate Gaussian distributions is interesting. The definition of the PGM distribution is certainly novel and I especially enjoyed the construction as it has the benefits of exact KL computation while enabling fast sampling. Finally, the overall flow and presentation of the paper at a high level are clear, but certain aspects could be improved to increase readability.

**Weaknesses:**

There are a few weaknesses in the paper which I outline below:

**Novelty:**

1. While the use of the PGM distribution is novel in a VAE setting the use of the Poincare Half plane model due to its enhanced numerical stability properties is not. In fact, this paper does not cite a key work in this space "Representing Hyperbolic Space Accurately using
Multi-Component Floats" Tao and De Sa 2021. I believe their usage of geometry shares all the same benefits of numerical stability (and perhaps even more) than the one presented here. I think this paper should attempt to implement a VAE with the Poincare Half-plane latent space as proposed in this paper for a fair comparison between approaches.

2. The authors comment that current hyperbolic normal distributions suffer from a variety of drawbacks such as numerical instability, lack of a closed form KL, and high computational cost in sampling. However, these criticisms seem mostly towards the Poincare and Riemannian Normal distribution. The wrapped normal does not have high sampling costs for instance, so the statement should be refined a bit. Furthermore, other hyperbolic distributions can be defined on higher dimensional hyperbolic spaces without resorting to product spaces. This is a limitation of the Poincare half-plane model itself, so I feel there should be more commentary about this and how it may affect experiments.

**Presentation weaknesses:**

1. The preliminaries section could be significantly improved. The introduction to Hyperbolic geometry is non-standard and the authors could lean on prior work to improve the exposition.
2. The definition of the Poincare half-plane model is insufficient here. You need to be much more concrete in mathematical terms on how it is defined. Things that one could add is what is the exact form of standard operations using this geometry, e.g. distance, exp map, log map, etc ... Furthermore, a visualization could improve readability.

**Experimental weaknesses:**

1. The experiments section has a few weaknesses. First, it seems that the bolded numbers in the tables have std's that overlap with other table entries. In this case, both table entries should be bolded.
2. I believe it is important to test this approach on actual tree-like datasets rather than image datasets. These include the Branching Diffusion Process dataset found in Nagano et. al 2019, as well as the lobster graphs dataset in Bose et. al 2020. A hyperbolic VAE should show more visual benefits in these settings.
3. Please include numerical error analysis in the same vein as Tao and De Sa 2021 Fig 1 and Fig 2 to substantiate the claims that the proposed approach has less numerical instability.





**Minor:**

1. The authors say "the tree-structured data can be embedded with arbitrary low distortion in hyperbolic space". This fact is only true for 2D hyperbolic spaces. The cited paper does not extend this to higher dimensions

**Questions:**

1. Past hyperbolic VAE's have made the curvature constant learnable which leads to improvements. Can this also be leveraged here?
2. In Table 1. the authors mark the HWN as not geometry aware. Can the authors explain why this is the case? The HWN explicitly uses the metric in all of its computations.
3. Similarly, the HWN has been used in VAE settings, in which the case KL divergence is needed and is not difficult to calculate. Can the authors explain why they feel this is not the case?

**Limitations:**

Yes.

---

> ### Author Rebuttal · Authors · 2023-08-09
>
> We sincerely thank the careful comments for the novelty of our method. We address the questions about the i) novelty on numerical stability, ii) drawbacks of HWN, and other minors.
>
> ---
>
> **Q1) PGM normal is another novel way to stabilize hyperbolic distribution.**
>
> We thank for introducing a work related to the stability of hyperbolic space. Although [1] tackles the stability issue using the Poincare half-plane model similar to our approach, adopting the Poincare half-plane model to the previous hyperbolic distributions is difficult, and PGM normal resolves the stability issue in a different direction.
>
> As suggested, we provide a machine error analysis on the models of hyperbolic space at section A of the general response. The results reveal that the Poincare half-plane model with MCF can be a better choice for HWN and Riemannian normal as suggested. However, it is hard to directly apply the proposed methods from [1] to the baseline distributions. Firstly, the computation of the normalizing constant and density function is necessary, and another new rejection sampling algorithm is also required for Riemannian normal. Achieving such technical contributions requires effort as done in previous work [2, 4] and is another interesting future research direction but out of the scope of our work. Secondly, we need to propose a bespoke architecture to apply MCF to VAE. In general, encoders in hyperbolic VAEs consist of Euclidean layers followed by a transformation layer to hyperbolic space. To apply MCF, we need to partition the encoder output to the components of MCF. This requires a carefully designed encoder layer and a partitioning algorithm because if the output of the encoder is too large, the output can be numerically unstable once transported to the Poincare half-plane.
>
> We finally emphasize that PGM normal relieves numerical instability in a different manner, not because of the adoptation of the Poincare half-plane model. Remind that the instability comes when the Riemannian operation is applied over the sampled points that do not belong to the manifold. The sampling and KL divergence computation of PGM normal would not suffer from instability since we can sample $\sigma$ in the log space, i.e., sample $\log \sigma$, which is much more stable than sampling $\sigma$ directly. Consequently, the KL computation can be done with $\log\sigma$. Furthermore, the log-space of the Poincare half-plane model is the same as the Euclidean space, so we do not need any exponential and log mappings. We also empirically show that PGM normal successfully runs with 16 bit precision in the RL task because of the simplified sampling and KL computation without any floating point trick. For more details, see the code density_estimation/distributions/PGMNormal/distribution.py and Appendix F.
>
> ---
>
> **Q2) The drawbacks of hyperbolic wrapped normal distribution.**
>
> We respectfully disagree that the mentioned drawbacks only appear in the Poincare normal.
>
> Firstly, HWN suffers from numerical instability due to the log map of the Lorentz model. As shown in Figure 1, the points of the Lorentz model escape from the manifold earlier than the Poincare half-plane model. Hence, the input value of the arccosh of the log map can be less than one, which is out of the domain and causes numerical instability.
>
> Secondly, the sampling speed is slow. As shown in Table 2, the sampling speed of HWN is 8.39x slower than the PGM normal due to additional mappings between the Euclidean space and the Lorentz model. This also affects the speed of the KL divergence computation which is approximated with Monte Carlo sampling. This eventually slows the overall training time of HWN as shown in Table 3 of the main paper.
>
> Lastly, the absence of KL divergence makes it hard to utilize complex covariance structure. Table 11 of Appendix D.3 of [3] shows that when using the KL divergence of non-diagonal covariance HWNs as the metric for word embedding, an extensive number of samples are required for stable training. This is not an issue when exploiting simple priors such as zero mean and identity covariance used in previous VAE. However, in the circumstances which require complex prior, e.g., conditional generation and world model of RL, it becomes problematic.
>
> ---
>
> **Q3)  Product space vs non-product space.**
>
> One of the advantages of non-product space is that we can adopt complex covariance structures for the distribution. While Poincare normal and PGM normal only allow isotropic covariance, HWN can take advantage of complex covariance structure. However, as mentioned in Q2, the absence of KL divergence bothers the usage of complex covariance. So all three distributions can take the advantage in limited cases. Also, the instability increases as seen in Appendix E.
>
> With the product space, by setting the curvature for each space differently, we can express a complicated manifold structure in a different way.  Although we cannot model the complex covariance of product space, we have a more flexible manifold structure.
>
> ---
>
> **Q4) Experiment on tree-like dataset.**
>
> We appreciate suggestions for new dataset. We train the VAEs on synthetic binary tree [2] and compare the density estimation performance. Table 1 and Figure 2 show that the hyperbolic VAEs outperform the Euclidean baseline in both the performance and the visualization.
>
> ---
>
> **Q5) Learnable curvature.**
>
> Please see section B of the general response.
>
> ---
>
> **Q6) Geometry of HWN.**
>
> Please check section C of the general response.
>
> ---
>
> **References**
>
> [1] Yu and De Sa, Representing Hyperbolic Space Accurately using Multi-Component Floats, NeurIPS 2021
>
> [2] Nagano et al., A Wrapped Normal Distribution on Hyperbolic Space for Gradient-Based Learning, ICML 2019
>
> [3] Cho et al., A Rotated Hyperbolic Wrapped Normal Distribution for Hierarchical Representation Learning, NeurIPS 2022
>
> [4] Mathieu et al., Continuous Hierarchical Representations with Poincaré Variational Auto-Encoders, NeurIPS 2022

---

> > ### Comment · Reviewer_bHMn · 2023-08-17
> > **Re:Rebuttal**
> >
> > I thank the authors for their rebuttal. I appreciate the new experiments provided in the 1-page PDF.
> >
> > Q1.
> >
> > The numerical stability of the proposed PGM model is not in doubt. The new Fig 1 shows this and should be included in the paper.
> >
> > > Firstly, the computation of the normalizing constant and density function is necessary
> >
> > For Wrapped distribution in the Lorentz model, this is known to us. Did you mean using a wrapped distribution in the PGM model?
> >
> > Q2.
> > > HWN suffers from numerical instability due to the log map of the Lorentz model
> > This is true. In practice, it is customary to clamp to an absolute value of $\approx 40$ as $\approx e^{40}$ is within floating point precision. Can you see a degradation in performance after clamping in your experiments for the rebuttal?
> >
> > > Secondly, the sampling speed is slow. As shown in Table 2, the sampling speed of HWN is 8.39x slower than the PGM normal due to additional mappings between the Euclidean space and the Lorentz model.
> >
> > What do you mean here? The projection is trivial. You sample a point in $$\mathbb{R}^{n-1}$$ and prepend a $0$ which makes it a point on the tangent space at the identity which is on the Lorentz model. After that, you can parallel transport and exp map to the manifold at a different point. Are you using this projection? Because this is not expensive.
> >
> > > This also affects the speed of the KL divergence computation which is approximated with Monte Carlo sampling.
> >
> > This I don't buy is slow. In general, this is super fast. An analytic KL in this case is not a big benefit for two reasons 1.) The main benefits of hyperbolic space are in low dimensional latents and for this is not a bottleneck 2.) In non-Gaussian VAEs, we use Monte Carlo all the time with no problems.
> >
> > Q3.
> >
> > I appreciate the answer but my original concern still remains.
> >
> > At the moment I am still unsure and need the authors to clarify my questions above. Based on these clarifications I may adjust my score.

---

> > > ### Author Response · Authors · 2023-08-17
> > > **Re:Re:Rebuttal**
> > >
> > > We appreciate participating in the discussion. We address the remaining concerns in the following response.
> > >
> > > ---
> > >
> > > **Q1. Further details for the HWN on the Poincare half-plane model.**
> > >
> > > We thank for the suggestion of quantitative analysis on numerical stability which strengthens our work. We will add Figure 1 in the future revision.
> > >
> > > We would like to clarify our understanding about the original review first. We thought that [1] shows that *the Poincare half-plane model* is a good choice for numerical stability *because* it is implemented on the Poincare half-plane model. Hence, the other baseline distributions, e.g, HWN and Poincare normal, *should be implemented on the Poincare half-plane model* for fair comparison on numerical stability. And this is why we answered “the computation of the normalizing constant and density function is necessary”.
> > >
> > > If this is the suggested direction of the original review (correct us if we misunderstood), we need to think about the details of the implementation. Let’s consider the HWN in the Poincare half-plane model. The first way to implement HWN on the half-plane is by transporting the sample of the HWN defined on the Lorentz model via isometry. However, the MCF of [1] is not validated on the Lorentz model and the isometries. The transformation also requires additional computation of the density through the change of variables. The second way is to re-define HWN on the Poincare half-plane model as done in [2, 4], as we mentioned in the previous rebuttal. The computation of density function and normalizing constant should be newly done in this case because the equations of the Riemannian operations change.
> > >
> > > ---
> > >
> > > **Q2. Results on clipped HWN.**
> > >
> > > We conduct an experiment on the effect of clipping in HWN. Before moving the encoder output to the Lorentz model to decide the mean value of the HWN, we clip the norm of the encoder output with the suggested bound. In some settings, there is a slight degradation but the confidence intervals overlap.
> > >
> > > ||CUB|Food101|Oxford102|
> > > |---|---|---|---|
> > > |latent dimension||||
> > > |50|$991.37_{\pm 2.83}$ | $1298.80_{\pm 8.74}$|$1298.75_{\pm 1.71}$|
> > > |60|$971.80_{\pm 2.83}$|$1228.73_{\pm 6.81}$|$1255.94_{\pm 2.27}$|
> > > |70|$954.00_{\pm 3.63}$|$1165.21_{\pm 4.57}$|$1232.40_{\pm 2.31}$|
> > >
> > > ---
> > >
> > > **Q3. Sampling speed of HWN.**
> > >
> > > We would like to emphasize the computation of MC-based KL is *relatively* slower than our closed-form solution, but in practice, as you suggested, this would not be a great problem. Here, we wanted to highlight the empirical difference between two methods but tone the presentation down in a future revision.
> > >
> > > We provide a detailed analysis on the sampling speed of the HWN and PGM normal. Both the HWN and PGM normal starts with sampling from the Euclidean normal. The HWN then transports the Euclidean sample to the Lorentz model via the projection mentioned in the re-response. The PGM normal finishes the sampling by concatenating the Euclidean sample with the Gamma distribution sample. So the difference between the two sampling procedures is the projection and the Gamma distribution sampling. Although there is no obvious difference between the two methods, the empirical results reveal that the Gamma distribution sampling is faster than the projection. However, in practice, we agree that the gap would not be noticeable during the training.
> > >
> > > We finally summarize our findings on the training speed of the HWN. Although the additional projection is just a combination of arithmetic operations and seems to have less effect on training VAE, we observe that there exists *a relative difference in the sampling speed* between PGM normal. Furthermore, while hyperbolic VAEs benefit more in low latent dimensions, the batch size is another important factor that affects the training speed. The below table shows that the training time does not vary on the latent dimension, while the relative difference is preserved (the absolute values differ from the paper due to different GPU).
> > >
> > > ||$\mathcal{E}$-VAE|$\mathcal{L}$-VAE|$\mathcal{P}$-VAE|GM-VAE|
> > > |---|---|---|---|---|
> > > |latent dimension||||
> > > |2|16.78|27.54|36.55|17.39|
> > > |4|16.42|28.74|38.15|17.55|
> > > |8|16.75|28.52|38.58|17.30|

---

### Official Review · Reviewer_ACLv · 2023-07-05

**Soundness:** 2 fair
**Presentation:** 3 good
**Contribution:** 2 fair
**Rating:** 5
**Confidence:** 3

**Summary:**

In this paper, the authors propose a novel Gaussian manifold generative autoencoder (GM-VAE) whose latent space is a statistical manifold formed by univariate Gaussian distributions. They also introduce a pseudo-Gaussian manifold normal distribution on the latent, which is easy to sample from and improves training stability due to a closed form of KL divergence. The experiments conducted show that GM-VAE outperforms the considered state-of-the-art baselines in the density estimation task on a few image datasets, and gives comparable results to DreamerV2 in the model-based RL task.

**Strengths:**

1. The proposed GM-VAE model is a non-trivial improvement of the VAE architecture with a solid theoretical background.

2. Experimental results on the CUB, Food101, Oxford 102, and (partially) Atari2600 Brekout datasets demonstrate the superiority of GM-VAE over its competitors ($\mathcal{E}$-VAE, $\mathcal{L}$-VAE, and $\mathcal{P}$-VAE).

3. The proposed model has superior computational performance concerning the other hyperbolic VAEs considered.

4. The paper is well written.

**Weaknesses:**

1. The experimental results of the GM-VAE in the model-based RL task are only comparable to those of competitors.

2. Rather small-scale image datasets were used in experiments.

**Questions:**

1. It would be interesting (and thus influence my opinion) to see the results of experiments performed on large-scale datasets (like CelebA). Also, the FID score would be reported (besides NLL).

2. Minor comments:

Tab. 2 caption: N/A appears only in the standard deviation (or perhaps confidence interval?) indices.

**Limitations:**

Generally yes, but for clarity I would suggest summarizing the limitations in a separate paragraph.

---

> ### Author Rebuttal · Authors · 2023-08-09
>
> We acknowledge with sincere help for improving our work. We provide the answers for the questions about i) the comparable results in model-based RL and ii) large-scale datasets.
>
> ---
>
> **Q1) The experimental results of the GM-VAE in the model-based RL task are only comparable to those of competitors.**
>
> We agree that the results are comparable. However, to the best of our knowledge, it is the first time in model-based RL to successfully run and achieve comparable results to the previous baseline with the hyperbolic distribution. We attempt to provide a new-axis to the RL community by applying hyperbolic space for model-based RL algorithm, where previous work is focusing on model-free algorithms [1].
>
> ---
>
>
> **Q2) Rather small-scale image datasets were used in experiments. It would be interesting (and thus influence my opinion) to see the results of experiments performed on large-scale datasets (like CelebA). Also, the FID score would be reported (besides NLL).**
>
> Due to the short rebuttal period, we could not conduct an experiment with datasets with larger images. However, we believe that the results from the RL environment, which we train with sequences of 84x84 images of length 50, implies that our method can benefit on larger-scale datasets.
>
> ---
>
> **Q3) Tab. 2 caption: N/A appears only in the standard deviation (or perhaps confidence interval?) indices.**
>
> Sorry for the confusion. The N/A appears only in the confidence interval as mentioned in the question. Instead, in the experiment results of non-product space in Appendix E, we can observe the N/A also at the mean values.
>
> ---
>
> **References**
>
> [1] Cetin et al., Hyperbolic Deep Reinforcement Learning, ICLR 2023

---

> > ### Comment · Reviewer_ACLv · 2023-08-16
> > **Thank you for the response**
> >
> > Thank you for the effort in preparing the rebuttal. I am open to updating my rating depending on the conclusions of the Reviewers-AC discussion phase.

---

> > > ### Author Response · Authors · 2023-08-17
> > > **Response to reviewer ACLv**
> > >
> > > We thank for responding to the rebuttal and being open to updating the score. Feel free to ask any questions in the remaining discussion period!

---

### Official Review · Reviewer_1SSZ · 2023-07-06

**Soundness:** 3 good
**Presentation:** 3 good
**Contribution:** 2 fair
**Rating:** 5
**Confidence:** 3

**Summary:**

Some VAE methods have been proposed to model the latent space as a hyperbolic space for modeling the structured representation of data. This paper proposes a hyperbolic distribution by using KL divergence to measure the geodesic distribution of Poincare normal, by using KL divergence as the local approximation of the geodesic distance. It makes the proposed method easy to sample and improve the numerical stability. Experiments on image density estimation and model-based RL are conducted.

**Strengths:**

+ The proposed method is with a clear motivation and relies on a simple assumption (in Eq. (2)) to make the analytic operations, which is straightforward.
+ Experiments are conducted on two different applications.
+ As shown in the reported results, the proposed method can perform better than or comparable with the related works.


**Weaknesses:**

- It is not clearly studied and discussed how the proposed pseudo Gaussian manifold normal distribution can model the hyperbolic distribution. Specifically, it is unclear how the introduced approximation may influence the modeling of the manifold. Although tractable KL is favorable, how the approximation introduced by the proposed method may influence the modeling capability?
- The authors may visualize the manifold of the proposed method and compare that with other methods, such as [27].
- The sampling results of the VAE models should be visualized and compared with previous methods, such as [27] and [25], especially considering that numerical improvements shown in tables are insignificant.
- As the claimed main benefit of the proposed method is numerical stability, the analysis and validation of the stability should be strengthened. How and why the previous works may introduce instability? How can the stability of the methods be measured and validated with empirical studies, numerical evaluation, and visualization?


**Questions:**

Please check other questions and suggestions pointed out with the weakness points.

**Limitations:**

The limitations are not explicitly discussed. The authors may check the weak points and consider discussing the potential limitation of the proposed method, such as the potential influence on the modeling capability that may be caused by the approximation.

---

> ### Author Rebuttal · Authors · 2023-08-09
>
> We thank for the sincere feedback. We address the issues about i) the modeling capability with local approximation, ii) visualization of our method and iii) analysis on numerical stability.
>
> ---
>
> **Q1)  It is not clearly studied and discussed how the proposed pseudo Gaussian manifold normal distribution can model the hyperbolic distribution. Specifically, it is unclear how the introduced approximation may influence the modeling of the manifold. Although tractable KL is favorable, how the approximation introduced by the proposed method may influence the modeling capability?**
>
> One of the important properties of hyperbolic space is that the pairwise distances of a tree can be perfectly embedded with the hyperbolic distance function. This enables deep neural networks to learn hierarchical representations by incorporating hyperbolic space as the latent space.
>
> In previous work, it is shown that KL divergence, the approximation of the hyperbolic distance function, can embed tree-structured data similar to the hyperbolic distance function [1, 2]. Based on the results, we believe that the approximation would not harm the capability of modeling the hierarchical structure.
>
> ---
>
> **Q2) The authors may visualize the manifold of the proposed method and compare that with other methods, such as [27].  The sampling results of the VAE models should be visualized and compared with previous methods, such as [27] and [25].**
>
> We add the density plots of the PGM normal at Figure 3. We run an experiment on tree-structured data to emphasize the visual benefits of hyperbolic VAEs. We train the models on a synthetic binary tree [2] and compare the performance in a density estimation task. Figure 2 describes the visualization of the learned latents.
>
> ---
>
> **Q3) As the claimed main benefit of the proposed method is numerical stability, the analysis and validation of the stability should be strengthened. How and why the previous works may introduce instability? How can the stability of the methods be measured and validated with empirical studies, numerical evaluation, and visualization?**
>
> We provide further explanation on the numerical instability of hyperbolic distributions and quantitative analysis.
>
> We first analyze the causes of numerical instability of hyperbolic space with different models. The numerical issues of hyperbolic distributions occur when the mean or the sampled point does not belong to the manifold due to the machine precision error. For example, the denominator of the distance of the Poincare disk model becomes zero when the point is placed on the border of the disk. In the Lorentz model, the input of the arccosh function of the log mapping can be less than one when the mean of the distribution is far from the origin and the sampled point is near the mean point, which is out of the domain. Theoretically, these issues should not happen, but we have observed the problems during the experiments due to the limited machine precision.
>
> These operations are essential in the sampling and KL divergence computation of the distributions. Inspired by the suggestion of reviewer bHMn, we conduct a quantitative analysis on numerical stability by following [1].  We compare the machine precision errors of the three models, the Poincare half-plane model, the Lorentz model, and the Poincare disk model which are used in PGM normal, HWN, and the Poincare normal with respectively. Figure 1 shows that the Poincare half-plane model, which is used in PGM normal, is more stable than the other two models in a wider regime.
>
> Furthermore, the sampling and KL divergence computation of PGM normal are free from the manifold of the Poincare half-plane model because we can express the operations with the log-space of the half-plane manifold, which is identical to Euclidean space. For more details, see the code of density_estimation/distributions/PGMNormal/distribution.py and Appendix F of the supplementary.
>
> We also empirically show that PGM normal is the only hyperbolic distribution that runs with 16 bit precision in the RL task while the other distributions fail and output NaN.
>
> ---
>
> **References**
>
> [1] Nagano et al., A Wrapped Normal Distribution on Hyperbolic Space for Gradient-Based Learning, ICML 2019
>
> [2] Cho et al., A Rotated Hyperbolic Wrapped Normal Distribution for Hierarchical Representation Learning, NeurIPS 2022

---

> > ### Comment · Reviewer_1SSZ · 2023-08-21
> >
> > Thanks for the response.
> > Can the authors produce the visualization of the manifold similar to (and in comparison with) Figure 8 in [27]??
> >
> >
> > As shown in Figure 2 in the response, the structure of the proposed method seems more distorted than other methods, which is not a good sign.

---

> > > ### Author Response · Authors · 2023-08-21
> > > **Response to reviewer 1SSZ**
> > >
> > > We thank for the response and additional feedback.
> > >
> > > We add the visualization of PGM normal at Figure 3 of our one-page PDF supplementary. Due to the page limit, we only differed the $\alpha, \beta$ value with fixed $\gamma$. We will add various visualizations of PGM normal in the future revision as suggested by the response.
> > >
> > > We apologize for occuring confusion due to the different visualization manifold. The latents of GM-VAE show unique behavior among the hyperbolic VAEs which looks distorted. We expect such behavior appears because of the different manifold, i.e., the Poincare half-plane model, but not because of the quality of embedding as reviewer concerns. We will add the visualization of the latents in the Poincare disk model as the other hyperbolic VAEs in the future revision. We conjecture that the visualization would be improved in a more intuitive manner, because the generation performance is better than $\mathcal{L}$-VAE as shown in Table 1.

---

### Author Rebuttal · Authors · 2023-08-09

We appreciate all reviewers for valuable comments which can improve our work. We reply to each reviewer with independent comments. We also provide supplementary material to complement the response to reviewers. The supplementary material consists of a PDF with figures and tables, and answers to frequently asked questions by the reviewers.

We organize the PDF with 4 figures and 3 tables. The figures are about the quantitative analysis on the numerical stability of the hyperbolic models [1SSZ, bHMn, Darj], the latent space of the hyperbolic VAEs [1SSZ, bHMn], the density plots of PGM normal [1SSZ], and the approximation performance between KL divergence and wrapped normal operation [Darj]. For the tables, we report the results of density estimation on synthetic binary tree [bHMn], detailed training speed analysis [bHMn], and the result of density estimation with learnable curvature GM-VAE  [bHMn, Darj].

In the next section, we answer the questions which are frequently asked by the reviewers.

---

**A. Quantitative analysis on the numerical instability [1SSZ, bHMn, Darj]**

We first analyze the causes of numerical instability of hyperbolic space with different models. The numerical issues of hyperbolic distributions occur when the mean or the sampled point does not belong to the manifold due to the machine precision error. For example, the denominator of the distance of the Poincare disk model becomes zero when the point is placed on the border of the disk. In the Lorentz model, the input of the arccosh function of the log mapping can be less than one, which is out of the domain. Theoretically, these issues should not happen, but we have observed the problems during the experiments due to the limited machine precision.

These operations are essential in the sampling and KL divergence computation of the distributions. Inspired by the suggestion of reviewer bHMn, we conduct a quantitative analysis on numerical stability by following [1].  We compare the machine precision errors of the three models, the Poincare half-plane model, the Lorentz model, and the Poincare disk model which are used in PGM normal, HWN, and the Poincare normal with respectively. Figure 1 shows that the Poincare half-plane model with multi-component floats (MCF) is more stable than the other models in a wider regime.

---

**B. Learnable curvature experiment [bHMn, Darj]**

PGM normal can leverage the learnable curvature. We test the PGM normal with learnable curvature on the density estimation task. We let all the curvatures learnable, and set the learning rate same as the VAE learning rate. We initialize the curvature to $-1$. Table 3 shows that there is no enhancement in the density estimation performance. Due to the short rebuttal period, we haven’t tested all possible configurations of learnable curvature, but we conjecture that carefully chosen learning rate and the initial value could improve the performances.


---

**C. Meaning of “Geometry” column [bHMn, Darj]**

The geometry column of Table 1 from the main paper means that the distribution represents the density of a sample point with the hyperbolic distance. Although the HWN utilizes operations with Riemannian metric in sampling, we thought that the resulting distance is dependent on the Euclidean distance, not the hyperbolic distance due to the normal distribution on the tangent space. However, we find that the density of HWN can be seen as a local approximation of the hyperbolic distance [2], so we will remove the table in future revision.

We conduct a study on the approximation performance of KL divergence and wrapped normal operation, which are used in PGM normal and HWN with respectively. The results are at Figure 4. While the KL divergence shows better approximation in points with small squared distance, the wrapped normal better approximate the squared distance in far points.

---

**References**

[1] Yu and De Sa, Representing Hyperbolic Space Accurately using Multi-Component Floats, NeurIPS 2021

[2] De Bortoli et al., Riemannian score-based generative modelling, NeurIPS 2022

---

### Decision · Program_Chairs · 2023-09-21

**Decision:**

Accept (poster)

**Comment:**

The paper introduces a novel method for VAEs with a latent space constructed using a hyperbolic distribution, modeled through the KL divergence to locally approximate the hyperbolic geodesic distance. The motivation is to improve numerical stability and make sampling easier. The experiments demonstrate its effectiveness on image datasets and model-based Reinforcement Learning (RL) benchmarks.

**Strengths:**
- Novel construction of a hyperbolic distribution, providing a unique way to form a hyperbolic latent space in a VAE setting.
- The method allows for straightforward analytic operations and is grounded in a clear motivation.
- The paper is well-written, with several reviewers noting its clarity and ease of understanding.
- Experimental results on multiple datasets showcase the potential superiority or comparability of this approach over its competitors.
- The factorization of the proposed distribution into known distributions is seen as an elegant feature.

**Weaknesses:**
- Several reviewers raised concerns about the clarity and thoroughness of the paper's exploration into the paper’s main benefit of numerical stability.
- Experimental results in the model-based RL task are only comparable to competitors, and some evaluations were performed on relatively small-scale image datasets.
- Some reviewers pointed out missing citations to relevant works.
- Presentation aspects, such as the introduction to Hyperbolic geometry and definition of certain models, were seen as lacking or non-standard.
- Reviewers noted that the authors could have delved deeper into potential limitations or challenges.

Given the aggregate of the reviews, the paper provides a technically solid contribution with a novel perspective on VAEs and hyperbolic spaces. However, there are concerns about the comprehensive demonstration of its benefits, especially numerical stability, and certain presentation aspects. A revision addressing the aforementioned weaknesses could bolster the paper's impact.